META-RESEARCH ARTICLE

# The characteristics of early-stage research into human genes are substantially different from subsequent research

Thomas Stoeger[1,2,3]*, Luís A. Nunes Amaral[1,2,4,5,6]*

**1** Department of Chemical and Biological Engineering, Northwestern University, Evanston, Illinois, United States of America, **2** Northwestern Institute on Complex Systems (NICO), Northwestern University, Evanston, Illinois, United States of America, **3** Center for Genetic Medicine, Northwestern University, Chicago, Illinois, United States of America, **4** Department of Molecular Bioscience, Northwestern University, Evanston, Illinois, United States of America, **5** Department of Physics and Astronomy, Northwestern University, Evanston, Illinois, United States of America, **6** Department of Medicine, Northwestern University School of Medicine, Chicago, Illinois, United States of America

* thomas.stoeger@northwestern.edu (TS); amaral@northwestern.edu (LANA)

**Data Availability Statement:** Data and code underlying figures can be obtained from https://doi.org/10.21985/n2-b5bm-3b17.

**Funding:** TS was supported by a grant of the National Institute on Aging (K99AG068544). LANA

## Abstract

Throughout the last 2 decades, several scholars observed that present day research into human genes rarely turns toward genes that had not already been extensively investigated in the past. Guided by hypotheses derived from studies of science and innovation, we present here a literature-wide data-driven meta-analysis to identify the specific scientific and organizational contexts that coincided with early-stage research into human genes throughout the past half century. We demonstrate that early-stage research into human genes differs in team size, citation impact, funding mechanisms, and publication outlet, but that generalized insights derived from studies of science and innovation only partially apply to early-stage research into human genes. Further, we demonstrate that, presently, genome biology accounts for most of the initial early-stage research, while subsequent early-stage research can engage other life sciences fields. We therefore anticipate that the specificity of our findings will enable scientists and policymakers to better promote early-stage research into human genes and increase overall innovation within the life sciences.

## Introduction

A stream of research [1–12] has now established that research into human genes currently investigates largely those genes that were already well studied in the past. Curiously, this narrow focus can only partially be explained by the relevance of these genes toward human health or physiology [9,13,14]. Rather, research suggests that this narrow focus can be primarily explained by physical, chemical, and biological properties of gene products that facilitated pregenomic investigations [11]. This insight is neither unique to genes nor surprising as science is a difficult and highly competitive endeavor, and researchers can therefore find themselves picking "the lowest-hanging fruit" within their chosen scientific domain [15,16].

was supported by grants of the National Science Foundation (1956338), Air Force Office of Scientific Research (FA9550-19-1-0354), National Institute of Allergy and Infectious Diseases (U19AI135964) and Simons Foundation (DMS-1764421), and a gift by John and Leslie McQuown. The funders had no role in study design, data collection and analysis, decision to publish, or preparation of the manuscript.

**Competing interests:** The authors have declared that no competing interests exist.

**Abbreviations:** COVID-19, Coronavirus Disease 2019; NCI, National Cancer Institute; NHGRI, National Human Genome Research Institute; NIA, National Institutes on Aging; NIGMS, National Institute of General Medical Sciences; NIH, National Institutes of Health.

Research directed toward human health has historically received high levels of financial and political support [17,18]. Along these lines, and irrespective of research into genes, the former president of the European Research Council postulated that research directions will only continue to receive stable public support if they align with societal goals [19]. We therefore anticipate that research toward little characterized genes could present researchers and policymakers with novel opportunities for aligning their efforts with societal goals—particularly if they concern genes implicated in human disease. For instance, we recently reported that over half of the human host genes relevant to Coronavirus Disease 2019 (COVID-19) lie outside of past patterns of scientific inquiry and therefore have remained uncharacterized in the months that followed the manifestation of COVID-19 as a global health burden [20]. Likewise, distinct research groups reported that genome-wide datasets identify genes likely to be important for cancer and neurological diseases, but that those genes have otherwise not been characterized [9,11,14].

The narrowness of the focus of life sciences research on human genes described above also raises practical concerns. For instance, current biological knowledge might be incomplete for many genes and processes [14], which may hinder the discovery of important associations. Indeed, gene ontology enrichment analyses could only reliably retrieve the well-known—since the 1960s [21]—association between cancer and the cell cycle from the late 2000s onward because only a small fraction of genes have been actively investigated. Further, studies of protein–protein interaction networks suggest that, physiologically, more essential genes also encode for proteins with more interaction partners. However, a recent study reported no evidence for such an association [22] after accounting for the narrow focus of the life sciences on a subset of genes through alternate experimentation or added normalization [23].

Partially echoing these societal and practical issues, several researchers and policymakers started efforts to promote research into little characterized genes. For instance, an international consortium, supported by the largest funding agencies of Europe and the United States, called for the establishment of a deep genome project to characterize all human genes and their orthologous murine genes [24]. Beyond such calls for actions, the largest funding source within the life sciences, the National Institutes of Health (NIH) of the US, already established a dedicated program to promote research into little characterized genes [10] and started to solicit funding opportunities that are restricted to genes that appear to be undercharacterized. The partial success of the underlying policies is already measurable as a recent bibliometric study found that during the most recent years, research into novel gene targets received a disproportionally high level of support from the NIH when compared to other sources of funding [25].

We believe that by understanding the factors that in the past have resulted in early-stage research on novel genes, we will be better placed to facilitate the design of initiatives and policies that support research into new or distinct sets of genes. While we can anticipate generalizing studies of science and innovation [26] to be helpful—particularly if those insights stem from studies based on life sciences research [27]—it remains unclear to which extent they offer appropriate guidance when studying genes and their encoded molecular products. Specifically, research into human protein-coding genes might be distinct from most other domains of scientific inquiry as scientists have been aware of the nearly complete [28] space of all possibly explorable entities after the advanced draft of the human genome sequence was published in 2001 [29–31]. Further, general insights into science and innovation may not suffice to identify the specific contexts that have enabled early-stage research into genes. This realization motivated us to revisit the past 50 years of research into human genes through discipline-wide datasets and delineate the contexts under which early-stage research into genes occurred. Recognizing the value and ideas of earlier studies into science and innovation, we will,

throughout our manuscript, introduce such earlier studies while pursuing hypotheses postulated or derived from them. Complementing these hypothesis-driven inquiries, we will also direct statistical approaches toward the available literature on human protein-coding genes.

Briefly, we will find through hypothesis-driven inquiries that early-stage research is produced by larger scientific teams, mentions more genes in titles or abstracts, and accrues more citations. However, its citation dynamics unexpectedly do not reveal a trade-off between high risks and high rewards. Nor is early-stage research separate from clinical trials. Further, we demonstrate that distinct phases of early-stage research affect citation patterns in the scientific literature differently. Through statistical analyses, we will then pinpoint the initial stage of early-stage research toward genome biology and reveal that subsequent early-stage research can be accounted for by a handful of medically important genes that have recently become investigated by scientific fields studying obesity and age-related neurological disease.

## Scientists rarely highlight novel gene targets

We begin our investigation by asking when the presently known and indexed genes were first highlighted in the title or abstract of a PubMed indexed publication. The construction of our database, and an evaluation of its error rates, is described in detail in the Methods section. Briefly, we find that the database should enable us to investigate early-stage research into human protein-coding genes during the past 50 years. Simultaneously, we would also like to caution readers of this manuscript that our investigation into the conditions surrounding early-stage research into human protein-coding during the past 50 years will adopt a selective and gene-centric perspective that is likely biased toward the perspective of present day US-based life scientists (see S1 Fig and "Study limitations" section).

We find that the number of newly highlighted genes increased most strongly between the mid-1990s and early 2000s (Fig 1A). By 2018, 86% of all genes had been highlighted at least once, and 76% of all genes had been mentioned by one of their names in the title of at least 1 publication (Fig 1A). Consistent with prior research [11], fewer than half (48%) of all genes are highlighted on any given year, and even fewer (36%) are mentioned in titles within a given year (S2 Fig). More optimistically, however, we can now state that most human genes have at least once been highlighted in a scientific publication.

Despite restricting our analysis toward research publications with genes highlighted in their titles or abstracts, we can confirm previous findings about the temporal dynamics of research into human genes during the last 2 decades [1–12] and extend them to 2018. First, the number of publications highlighting a given gene correlates strongly over time (Spearman: 0.79, S3A Fig). Second, the number of publications per gene during this 10-year window negatively correlates with the year in which a gene was first highlighted (Spearman: −0.59, S3B Fig). Third, starting in the late 1990s, only a vanishing fraction of gene–publication pairs published in a given year address newly (highlighted in the same year) or recently (highlighted within the past 5 years) highlighted genes (Fig 1B). Fourth, starting in the late 1990s, newly and recently highlighted genes became underrepresented relative to their absolute number (Fig 1C), even when accounting for the physiological importance of genes through their loss-of-function intolerance in human populations [32] (S4A Fig) or the occurrence of phenotypes in systematic murine mutagenesis experiments [33] (S4B and S4C Fig) or genome-wide association studies [34] (S4D Fig).

These results clearly demonstrate that during the last 2 decades, early-stage research into human genes has become a small fraction of the scientific inquiry into human genes [1–12]. We further anticipate that, going forward, the potential for early-stage research will continue to shrink. First, the number of human protein-coding genes is finite. Second, there already

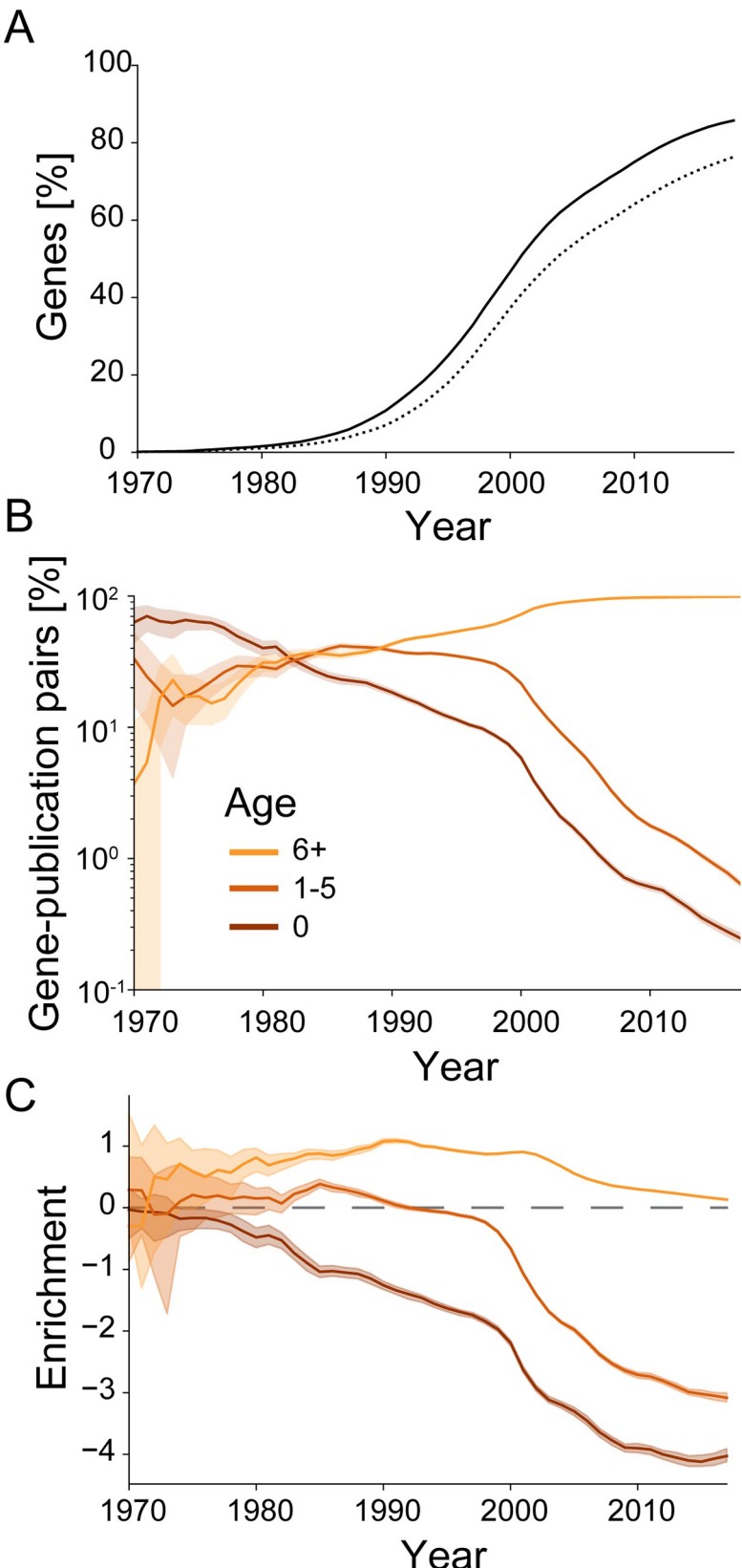

**Fig 1. During the 1990s, early-stage research into currently acknowledged genes became underrepresented. (A)**
Share of human protein-coding genes that have been highlighted (mentioned by name in title or abstract) until the
indicated year (solid) or mentioned by name in the title (dotted). **(B)** Share of all pairs of highlighted genes and
research publications that fall onto newly highlighted genes (0 years, brown) or recently initially highlighted genes (1 to
5 years, dark orange) or old gene targets (6 or more years, light orange). Shaded area indicates 95% confidence
intervals inferred by bootstrap. Year indicates center of a 3-year sliding window used for analysis. **(C)** Enrichment
(log$_2$ of ratio) of annual pairs of highlighted genes and research publications relative to the number of genes falling into
the categories presented in panel B. Shaded area indicates 95% confidence intervals inferred by bootstrap. Year
indicates center of a 3-year sliding window used for analysis. Combines data from MEDLINE, NCBI gene and
taxonomy information, gene2pubmed, and PubTator. For data underlying the figure, see https://doi.org/10.21985/
n2-b5bm-3b17.

appears to be a saturation of the annual number of newly highlighted genes (Fig 1A). Third, by
2018, the fraction of genes that have not yet been highlighted at least once has decreased to
only 2.2% of those genes that are under an extraordinary high evolutionary pressure in human
populations and therefore appear particularly important to human physiology or fecundity
(S2C Fig).

## Early-stage research into genes is distinctly produced and received

As innovation within the sciences has been extensively investigated through bibliometric prop-
erties of scientific publications [35], we will next use bibliometric properties to lead our inquiry
into the conditions surrounding early-stage research into human genes. We will test 2 main
hypotheses: Hypothesis 1: Early-stage research into genes is produced under distinctive condi-
tions from research into older genes (first highlighted 6 or more years prior to given publica-
tion) and hypothesis 2: Early-stage research into genes is distinctively received by the broader
scientific community.

We noted that even miniscule differences (e.g., 3% change in average number of authors)
could reach statistical significance at $p$-values <0.01 given the large number of publications
considered for statistical analysis. As such, we will generally refrain from reporting significance
values. Instead, we focus our text on effect sizes, while displaying bootstrapped confidence
intervals in the figures to directly convey the uncertainty surrounding our metrics.

To test the first hypothesis, we build on research suggesting that teams of different sizes
produce distinct types of innovation [36,37]. We find that until the early 2000s, the average
and median number of authors on early-stage publications closely matches the average and
median number of authors of publications highlighting only older genes (Fig 2A, S5A Fig).
From 2010 onward, we observe that early-stage research into genes is produced by much larger
teams than observed for teams authoring publications highlighting only older genes (Fig 2A,
S5A Fig). This increase also exceeds the increase anticipated from extrapolating from preced-
ing increases in mean and median team sizes (Fig 2A, S5A Fig). We thus conclude that early-
stage research into genes has recently become the purview of larger scientific teams.

To continue to test the first hypothesis, we turn from authors toward their manuscripts and
ask whether these manuscripts differ beyond the inclusion of novel or recent gene targets. Sup-
porting this view, we find that since the 1970s, the average number of genes highlighted in
publications that newly highlight a gene is nearly the double of the average number of genes
highlighted in publications that only highlight older genes (Fig 2B). Similarly, we observe the
median number of genes to have increased too (S5B Fig). We thus conclude that manuscripts
reporting on early-stage research into genes are characterized by larger scopes in terms of the
number of considered genes. Our findings on team sizes and the number of genes thus lead us
to conclude that our first hypothesis is correct and that early-stage research into genes is pro-
duced under conditions that differ from research on older gene targets.

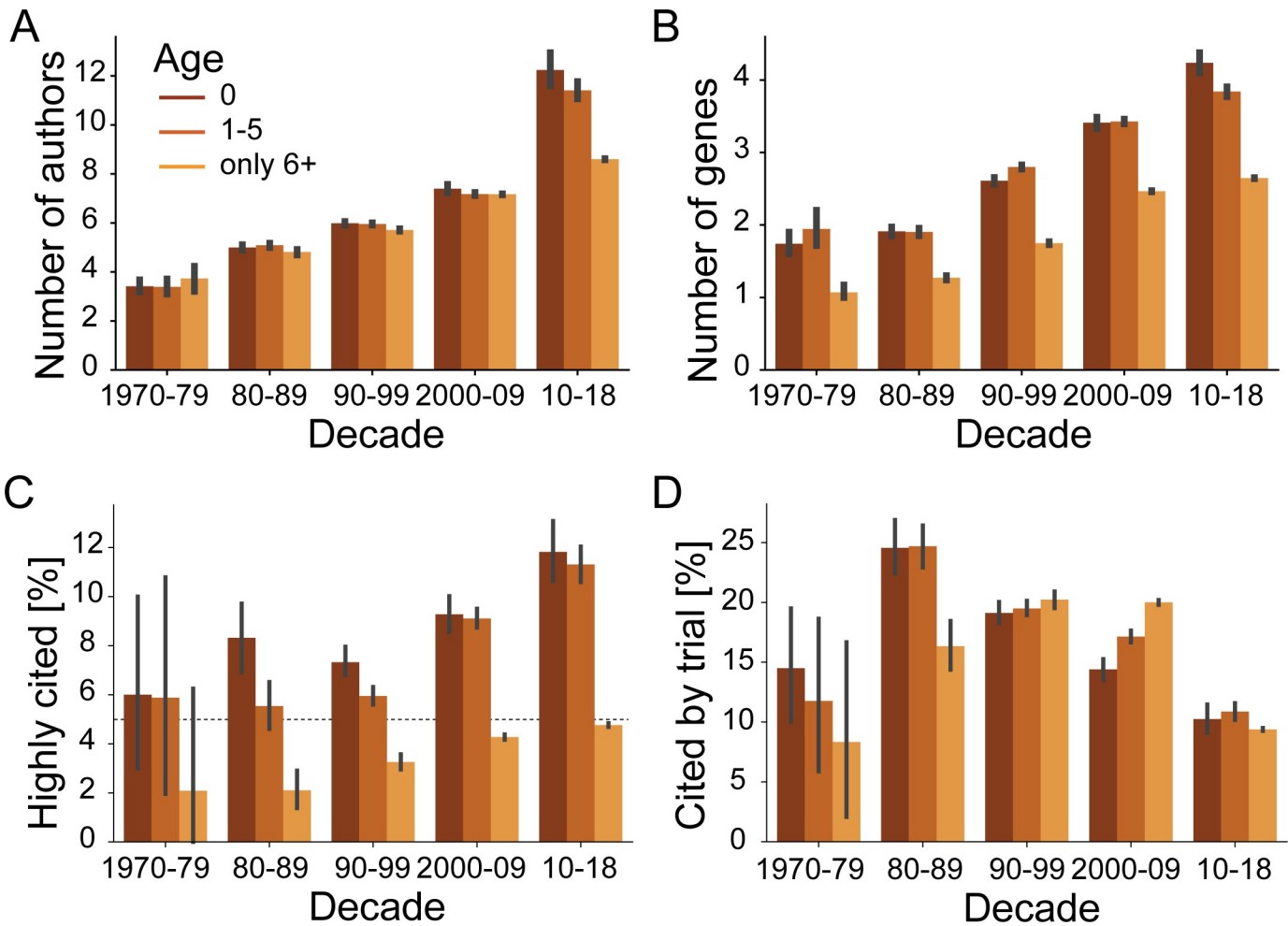

**Fig 2. Publications introducing newly highlighted genes are published by larger teams, highlight more genes in title and abstract, and are more likely to become highly cited.** (**A**) Average number of authors for publications highlighting at least 1 gene aggregated by age of highlighted gene and by decade. (**B**) Average number of highlighted genes per publication for publications highlighting at least 1 gene aggregated by age of highlighted gene and by decade. (**C**) Share of highly cited publications (among top 5% of year of publication) per decade. Dashed line shows 5% baseline. (**D**) Share of publications highlighting at least 1 gene aggregated by age of highlighted gene and by decade that are cited by at least 1 clinical trial. Error bars show 95% confidence intervals inferred by bootstrap. Gene ages at publication are grouped identically in all panels and as shown in legend of panel A. Combines data from MEDLINE, NCBI gene and taxonomy information, gene2pubmed, PubTator, and iCite. For data underlying the figure, see https://doi.org/10.21985/n2-b5bm-3b17.

To test the second hypothesis, we build on research that suggests that innovative research is rewarded with exceptional peer recognition—awards, prizes, and election to learned societies —and with increased number of citations [27,35]. Due to data availability to us, here, we only test here whether early-stage research into human genes is more likely to become highly cited. Confirming our hypothesis, we find that since the mid-2000s, the probability of an early-stage research publication becoming highly cited was double that of publications highlighting only older genes (Fig 2C). We can thus see that the scientific enterprise possesses mechanisms to recognize and reward innovative, early-stage research.

An increased probability for becoming highly cited could hint toward early-stage research generally being more cited. Alternatively, and consistent with the idea of novelty-pursuing research representing a trade-off between high risk and high reward, novelty-pursuing research was previously found to be accompanied by an increased variability in citation rates

—leaving some publications particularly frequently cited while other publications would be particularly infrequently cited [38]. Briefly, we find support for the former possibility of early-stage research generally receiving more citations (S6A Fig), whereas—except for 3 years around the turn of the millennium—publications on early-stage research have indistinguishable variability in citation rates (S6B Fig).

Because recent research highlights that the number of accrued citations yields an incomplete picture of the subsequent reception of scientific work [39,40], we consider an alternative metric for impact. In the context of the life sciences, a group led by George Santangelo at the NIH's Office of Portfolio Analysis recently suggested to place a special emphasis on citations by clinical trials [41]. We might anticipate that early-stage research is the furthest from the development of therapies and therefore has the lowest probability to be cited by at least 1 clinical trial [42]. Revealing unexpected nuance, we find this only to be the case for publications in the period 2000 to 2009. Intriguingly, we observe that 1980s publications reporting early-stage research publications were more likely to be cited by clinical trials than publications on old gene targets (Fig 2D). Similarly, only from the 1990s onward are publications on old gene targets cited sooner by clinical trials (S6C Fig). We conclude that the relation between early-stage research and applied research can change and that clinical trials acknowledge early-stage research.

Taking a broader view of the role of innovation within the sciences, one might consider how much new research publications uncouple subsequent scientific literature from preceding literature [36] Using the disruption index recently introduced by Funk and Owen-Smith [43], and popularized by Wu and colleagues [36], we observe that publications reporting on novel gene targets are only marginally (approximately 3 percentile points) more disruptive than publications reporting only on old gene targets (S6D Fig). Further, publications reporting on recent genes targets that had first been highlighted during the 5 preceding years are even approximately 15 percentile points less disruptive (S6D Fig). These results suggest that different phases of early-stage research affect the evolution of the scientific literature differently and align with the observation of research into human genes rarely turning toward novel gene targets [1–12].

Still, we can conclude that, in agreement with our second hypothesis, early-stage research into human genes is distinctively received by the broader scientific community. Interestingly, we find that early-stage research into genes follows some, but not all, of the characteristics that are associated in the literature with scientific innovation and novelty.

## Characteristics of publication outlets for early-stage research

Our preceding bibliometric analyses demonstrate that early-stage research into human genes is produced under distinct conditions and that it is distinctively received by fellow scientists. However, the preceding analysis does not identify the specific contexts under which early-stage research into genes is produced or whether these contexts change over time.

We begin our identification of contexts by inspecting scientific journals and focusing on the period 2010 to 2018. While editors working for scientific journals appear to personally value novelty [44], the peer-reviewing process within scientific journals had been postulated to limit novelty [45–47]. First, we focus on those journals that published research on novel gene targets (Fig 3A, S7 Fig). We find an overall trend (gray dots) suggesting that the number of research publications reporting on novel gene targets scales with the number of research publications studying old gene targets (Spearman: 0.49, S7 Fig). This means that, as a first approximation, early-stage research into novel gene targets tends to appear in journals that publish many papers on human genes.

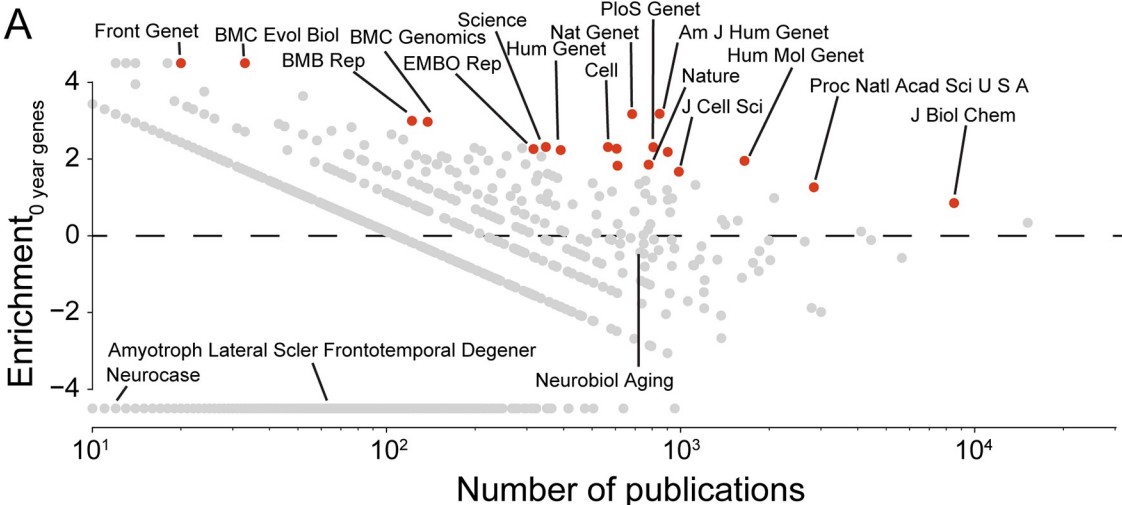

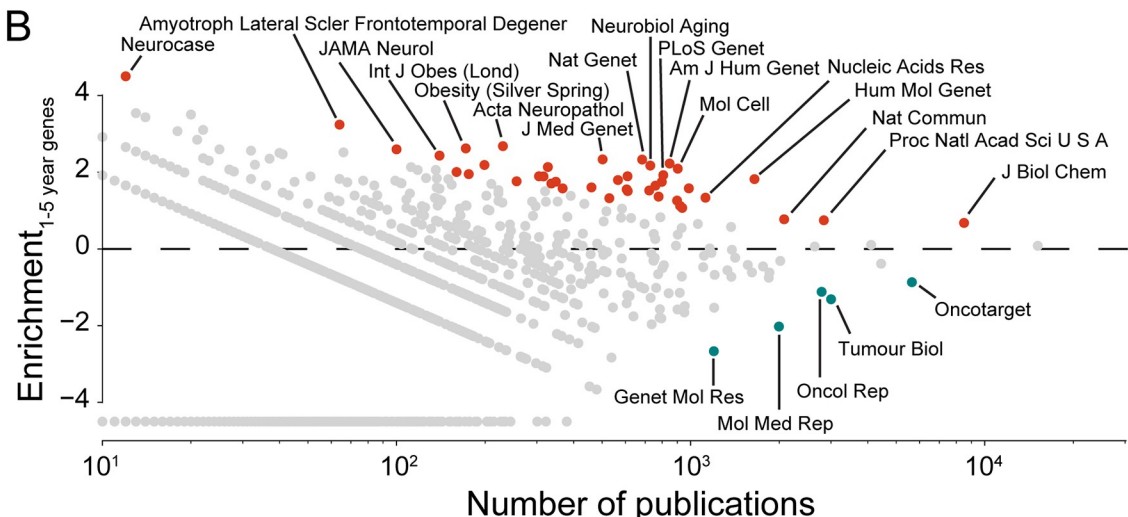

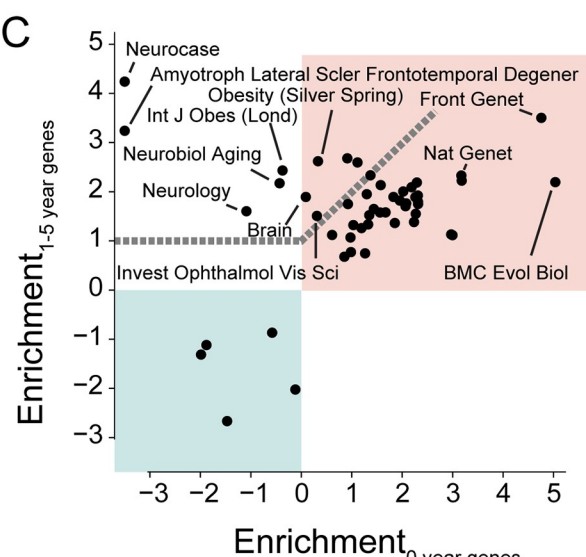

**Fig 3. Generalist journals and journals for genome research and age-related neuropathies associate with early-stage research in the period of 2010 to 2018. (A)** Enrichment ($\log_2$ of ratio) for publications with at least 1 new gene target (never highlighted in a preceding year; marked as 0-year genes). Each circle represents an individual journal; red (blue) circles indicate journals that significantly enrich (deplete) for publications on novel gene targets at Bonferroni multiple testing corrected Fisher exact test with $p < 0.01$. **(B)** Enrichment for publications on recent gene targets (first highlighted 1 to 5 years before). **(C)** Comparison of fold enrichment for journals that are significantly enriched or depleted for new gene targets (panel A) and/or recent gene targets (panel B). Note that for any given journal, the enrichment may not be statistically significant for both axes. Notably, the light cyan quadrant shows journals that are depleted from early-stage research. Note also that circles for the journals *Neurocase* and *Amyotroph Lateral Scler Frontotemporal Degener* are plotted "outside of the range of the axis" because they do not have any publication on new gene targets. Journals above dashed line enrich for recent gene targets at least twice as much as they do for novel gene targets. Combines data from MEDLINE, NCBI gene and taxonomy information, gene2pubmed, and PubTator. For data underlying the figure, see https://doi.org/10.21985/n2-b5bm-3b17.

Nonetheless, we find that some journals enrich for research publications on novel gene targets to an extent that is higher than anticipated by chance (red dots) (Fig 3A, S7 Fig). Manually inspecting their identity from the 1990s onward (S7 Fig), we believe to recognize 2 groups of journals. The first group consists of journals dedicated primarily to genome biology, whereas the second consists of journals that target an interdisciplinary audience and, in the cases of Nature, Science and PNAS, also publish research beyond the life sciences. We conclude that these 2 groups of journals are particularly effective at attracting innovative research into human genes.

Next, we focus on research publications on gene targets that were first highlighted in the 5 preceding years (Fig 3B, S7 Fig). We again find that some journals enrich for publications on early-stage research on gene targets to an extent that is higher than anticipated by chance. Again, many of those journals are directed toward genome biology, whereas a second group of journals have a greater disciplinary focus. Several of the journals in this second group relate to neurobiology and obesity (Fig 3B).

In contrast to the analysis for the enrichment for novel gene targets, when considering early-stage research, we find that few journals deplete for publications on early-stage research on gene targets to an extent that is higher than anticipated by chance (cyan dots) (Fig 3B, S7 Fig). Most seem to focus on cancer biology, but a noteworthy exception is the *Genetics and Molecular Research* journal (Genet Mol Res), which had been listed on Beall's list of potential predatory journals [48].

Comparing the extent to which journals enrich for newly highlighted versus recently highlighted genes, we find that journals dedicated to neurobiology and obesity are enriching for recently highlighted genes to a larger extent than their enrichment for newly highlighted genes (Fig 3C). We conclude that while there are strong similarities for the publishing contexts that surround newly and recently highlighted genes, some research fields might be more receptive toward publishing research publications on gene targets that had been identified recently. Expanding on the latter insight, we will return to research into genome biology, neurobiology, and obesity in later sections of this manuscript.

## Characteristics of institutional support for early-stage research

We round off our identification of specific contexts that may support early-stage research by turning toward institutions. We consider funding agencies, distinct funding mechanisms, and NIH-supported research organizations (Fig 4, S8–S11 Figs).

The contexts implicated through funding agencies resemble those presented among journals—with the support for early-stage research between 2010 and 2018 preferentially stemming from genome biology (NIH's National Human Genome Research Institute [NHGRI]—one of several different agencies within the NIH), generalist funding agencies (NIH's National

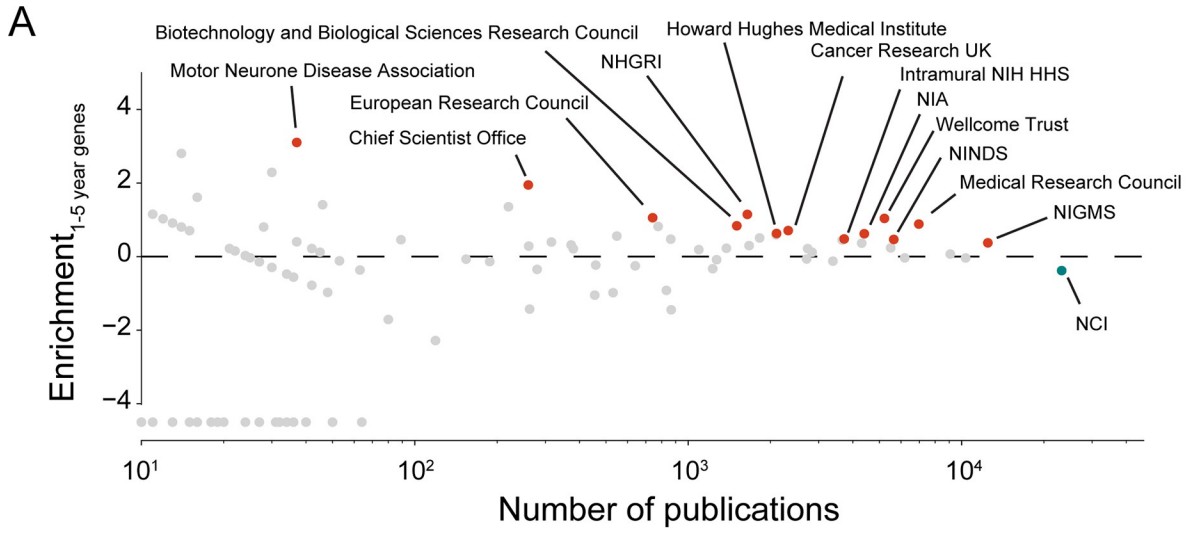

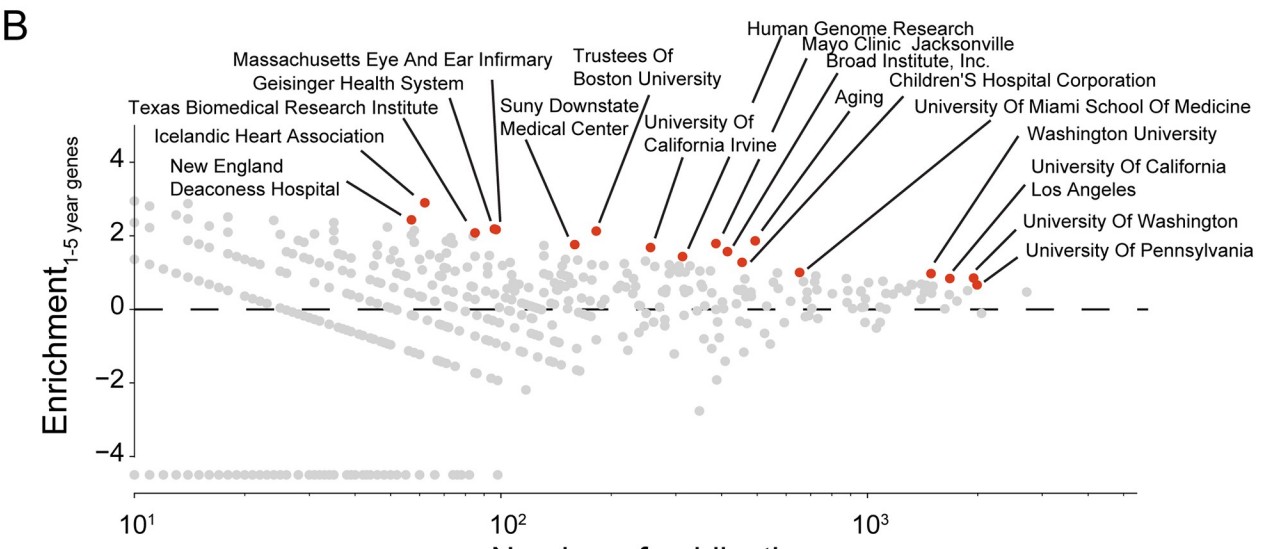

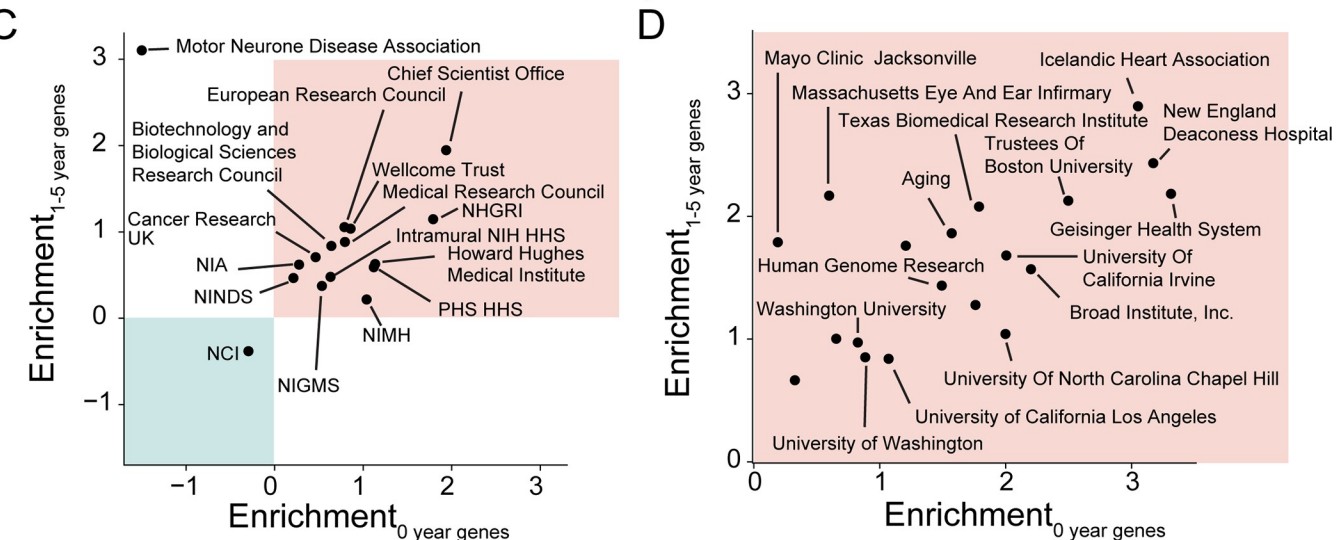

**Fig 4. Institutional support for early-stage research associates with genome biology in the period of 2010 to 2018. (A)** Enrichment (log$_2$ of ratio) for the support of specific funding agencies in publications highlighting at least 1 recent gene target (first highlighted 1 to 5 years before). Each circle represents an individual journal; red (blue) circles indicate journals that significantly enrich (deplete) for publications on novel gene targets at Bonferroni multiple testing corrected Fisher exact test with $p < 0.01$. **(B)** Enrichment for author affiliation to research organizations in publications highlighting at least 1 recent gene target. **(C)** Comparison of fold enrichment for funding agencies that are significantly enriched or depleted for new gene targets or recent gene targets. **(D)** Comparison of fold enrichment for research organizations that are significantly enriched or depleted for new gene targets or recent gene targets. Note that for any given funding agency or research organization, the enrichment may not be statistically significant for both axes. Notably, the light cyan quadrant shows funding agencies and research organizations that are depleted from early-stage research. Combines data from MEDLINE, NCBI gene and taxonomy information, gene2pubmed, PubTator, and ExPORTER. For data underlying the figure, see https://doi.org/10.21985/n2-b5bm-3b17.

Institute of General Medical Sciences [NIGMS], European Research Council, etc.), or funders of age-related neuropathies (Motor Neurone Disease Association and NIH's National Institutes on Aging [NIA]) (Fig 4A–4C, S8 Fig). In line with a prior study that reported that Howard Hughes Medical Institute Investigators may be more willing to change their research focus [27], we also find that publications supported by the Howard Hughes Medical Institute are enriched for early-stage research (Fig 4A–4C, S8 Fig). Finally, we also observe that publications authored by NIH intramural scientists are enriched for early-stage research.

We find that preferences of funding agencies toward early-stage research can be dynamic (S8 Fig). While the NIH's NHGRI and NIGMS have consistently supported researchers producing publications enriched for early-stage research, the NIH's National Cancer Institute (NCI) supported researchers producing publications enriched for early-stage research in the 1980s and 1990s, but since 2010 has been supporting researchers producing publications depleted for early-stage research (S8 Fig). This change contrasts with the funding patterns of Cancer Research UK, which has consistently supported researchers producing publications enriched for early-stage research (Fig 4B–4D, S8 Fig). This shows that the focus on early-stage research among domain-specific funding agencies can differ across countries and can change over time as those agencies reassess and reprioritize the areas of research they will pursue.

Funding agencies can allocate their funds through a plethora of distinct mechanisms, which vary in the scope of eligible scientists and research organizations, the duration of the support, amount of funding, and objective of the supported research. Because of data availability, we will again focus on the NIH, which informs on its different funding mechanisms through "activity codes." Overall, we find 112 different activity codes to have been used within 2010 and 2018, of which only 8 support publications that significantly enrich for early-stage research (S9 and S10A Figs). These activity codes also appear marginally more frequently deployed by those institutes of the NIH that disproportionally enrich for early-stage research (Mann–Whitney U: 0.05) (S10B Fig). Among these 8 activity codes, 3 relate to NIH's intramural research program (N01, ZIA, ZIB), 2 relate to research activities funded through contracts rather than research grants (U01, Z01), another relates to funding directed toward the career development of PostDocs (F32), and another is directed toward supporting high impact interdisciplinary science (RC2). The last activity code supporting publications enriching for early-stage research is the R01, the most common activity code (S9 Fig). While the enrichment is statistically significant for R01-type grants due to the very large number of observations, the magnitude of the enrichment is miniscule (6%). We thus conclude that only a minority of funding mechanisms preferentially supports early-stage research and that, essentially, only 2 of them correspond to funding that is allocated through extramural research grants.

Concluding our analysis of the research contexts under which early-stage research occurs, we investigate research organizations that are recipients of NIH grants (Fig 4B–4D, S11 Fig). The strongest enrichments for early-stage research are seen in a group of smaller research organizations—namely Geisinger Health Systems, the New England Deaconess Hospital, and

the Icelandic Heart Association. Manually inspecting their early-stage publications from 2010 onward, we see that each publication either describes a genome-wide association study, its verification, or a meta-analysis of association studies. Among the larger research organizations, we see (filed under 2 different names) the University of Washington, which historically played a prominent role in the Human Genome Project [49], and the Broad Institute, which, according to its mission statement, "was founded in 2004 to fulfill the promise of genomic medicine" [50]. Complementing above research organizations, we also see the intramural branch of the NIH's NHGRI and NIA. Our unbiased analysis of publication outlets, funding agencies, and research organizations thus all point toward genome biology as disproportionally promoting early-stage research into human genes.

## Genome biology accounts for most of the research into novel gene targets

Given the prominence of genome biology among scientific journals, funding agencies, and research organizations that produce or support publications that enrich for early-stage research, we next set out to quantify the extent to which the field of genome biology contributes to early-stage research into human genes. We classify publications as genome biology according to their title, abstract, and keyword sections as described in detail in our Methods section.

We observe that genome biology becomes identifiable as a field in the 1980s. Since the 2010s, it directly accounts for 49% of the early-stage publications on novel human genes (Fig 5A)—with this share having increased even further to 58% in 2018, the last year of our observation (S12 Fig). To understand how novel gene targets are first highlighted outside of publications on genome biology, we randomly select and manually review 20 such publications from 2015 and 2018 that were classified as not belonging to genome biology. We find that 12 of the 20 research publications use sequence similarities to other genes to select the gene which they would subsequently highlight in their own title or abstract (e.g., by using sequence similarity to define and/or find genes belonging to an acknowledged gene family), 2 build upon earlier work in mice, 1 identifies the new gene target through differential mass spectrometry, 1 identifies the new gene target through a functional cDNA screen, and 1 investigates the genes residing at a multigenic locus. Three research articles contain genes that were highlighted before but were missed by our pipeline (Fig 5B). We conclude that—extending beyond our definition of genome biology—several publications on new gene targets presently use information about genomes to identify their targets.

Previously, we observed that early-stage research into genes engages more authors, highlights more genes, and is more likely to receive a higher number of citations (Fig 2). To distinguish whether these bibliometric properties characterize early-stage research per se or just those falling under genome biology, we repeat the earlier analyses while distinguishing between genome biology publications and nongenome biology publications. We provide details about each decade and rank-based comparisons in S13 and S14 Figs, but focus our discussion here on the most recent period: 2010 and 2018.

We recognize that the previously noted increase in team size for early-stage research (Fig 2A) stems from genome biology (Fig 5C). By contrast, the previously noted increase in the number of highlighted genes (Fig 2B) also persists outside of genome biology (Fig 5D) but is twice as large for genome biology, while publications on old gene targets highlight the same number of genes irrespectively of whether they fall within genome biology or not (Fig 5D). The previously noted increased probability for early-stage research to become highly cited (Fig 2C) appears independent of genome biology (Fig 5E). However, genome biology publications on old gene targets are also more frequently highly cited (Fig 5E).

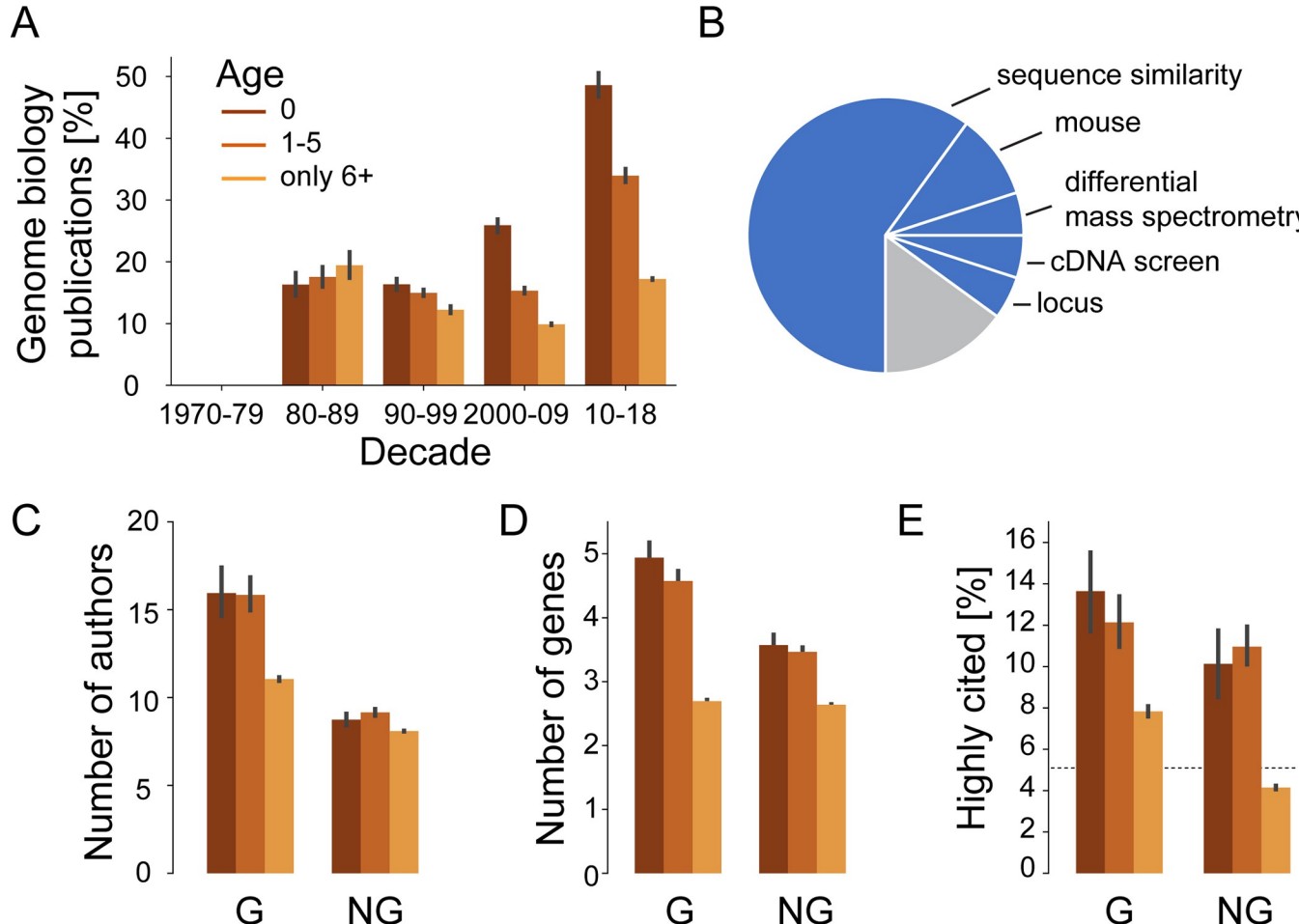

**Fig 5. Early-stage research is more frequent within genome biology publications. (A)** Share of publications highlighting at least 1 gene aggregated by age of highlighted gene and by decade that are categorized as genome biology. **(B)** Sources of novel gene target extracted manually for 20 randomly selected publications from 2015 to 2018 that highlight a new gene between 2015 and 2018 but were not categorized as genome biology. **(C)** Mean number of authors on 2010 to 2018 publications highlighting at least 1 gene aggregated by age of highlighted gene and field categorization ("G" for genome biology and "NG" for nongenome biology). **(D)** Mean number of highlighted genes for 2010 to 2018 publications highlighting at least 1 gene aggregated by age of highlighted gene and field categorization. **(E)** Share of highly cited 2010 to 2018 publications highlighting at least 1 gene aggregated by age of highlighted gene and field categorization. Gene ages at publication are grouped identically in all panels and as shown in legend of panel A. Error bars show 95% confidence intervals inferred by bootstrap. Dashed line shows 5% baseline. Combines data from MEDLINE, NCBI gene and taxonomy information, gene2pubmed, and PubTator. For data underlying the figure, see https://doi.org/10.21985/n2-b5bm-3b17.

We conclude that the bibliometric properties that characterize early-stage research are more pronounced within genome biology but—with the exception of team size—remain valid within nongenome biology publications.

We also noted that genome-wide association studies seem to be common among studies describing early-stage research. We thus perform bibliometric analyses that focus on genome research, but then distinguish between genome-wide association studies (including transcriptome-wide association studies) and other genome research. We find that the previously noted increase of in team size for early-stage research only holds to a notable extent for genome-wide association studies (S15 Fig). We also find that early-stage research on genome research outside of genome-wide association studies shows an increase that is smaller than the one that we had observed before for genome research (Fig 5D, S15 Fig) and that the increase of the probability to become highly cited of approximately 4% closely matches the probability seen for the

entire genome research (Fig 5E, S16 Fig). By contrast, the latter increase is small when compared to genome-wide association studies, where the chance to be among the 5% most cited publications of a year nearly doubles from approximately 30% for research on old gene targets to around approximately 55% for research on novel gene targets (S16 Fig). We conclude that some of our insights about the intersection between genome research and early-stage research could stem from a very specific domain of genome research. In line with the overall theme of our work, this further indicates the necessity and possibility to identify the specific contexts behind early-stage research as some beliefs about innovation might not generalize.

## Domain-specific research into a handful of genes contributes to early-stage research

While approaches used within genome biology are well positioned to identify novel gene targets [29] (Fig 5A), we previously observed that early-stage research on recent gene targets differs from early-stage research on novel gene targets by further extending to journals dedicated to neurobiology and obesity (Fig 3). This prompts us to ask whether recent gene targets that are of interest to specific domains of biology become the focus of greater interest than other recent gene targets.

We again focus here on the period of 2010 to 2018 for timeliness, but similar analyses for other periods are included in S17 Fig. During this time period, we find that 3,943 newly highlighted genes could have been investigated as recent gene targets (Fig 1A). A total of 1,656 (42%) were never highlighted during the same time period, and 885 (22%) are highlighted only once more. By contrast, and supporting our hypothesis, we find that a small number of recent gene targets become the focus of dramatically greater interest. Specifically, the 40 top ranked genes represent a little over 1% of the newly highlighted genes, but 21% of all unique pairs of recent gene targets and publications (Fig 6A, S17 Fig).

This small subset of top ranked recently highlighted genes defies the general trend of biomedical research rarely turning to new gene targets (Fig 1B and 1C, S3 Fig) [14]. By contrast, this unusual group of 40 genes (0.2% of all 19,171 human protein-coding genes) have already been highlighted in more publications than most human genes (Fig 6B). The most frequently highlighted genes within this group are FTO, C9orf72, and PALB2. By 2018, they had been highlighted in the title and abstract of more publications than 98%, 97%, and 94%, respectively, of protein-coding genes. These 3 genes strongly associate with obesity [51], dementia [52,53], and cancer [54,55], respectively. Interestingly, while FTO and C9orf72 were initially discovered through genome-wide association studies (and thus through genome biology), PALB2 was initially discovered through its biochemical interaction with BRCA2, a prominent cancer gene [56]. These results highlight the potential biological importance of novel gene targets and the potential to become heavily investigated within a few years.

Finally, we return to our observation of neurological and obesity-related journals enriching for early-stage gene targets to an extent that exceeds their enrichment on novel gene targets (10 journals above dashed line, Fig 3C). Plotting the share of their targets which fall onto the 40 top ranked genes, we find that it ranges between 32% and 100% and thus significantly exceeds share observed among other journals that enrich for recent gene targets (Fig 6C). To identify whether these journals focus on similar gene targets, we next cluster these journals according to the subset (15) of the 40 top ranked genes that have been highlighted by them at least once (Fig 6D). We recognize 3 groups. Seven journals—all dedicated to neurology—primarily include C9orf72 as a recent gene target. Two journals about obesity primarily include FTO as a research target, and to a lesser extent, 5 other genes linked to obesity [57]. Finally, one cluster is formed by a single journal about vision, which primarily highlights ARMS2—a

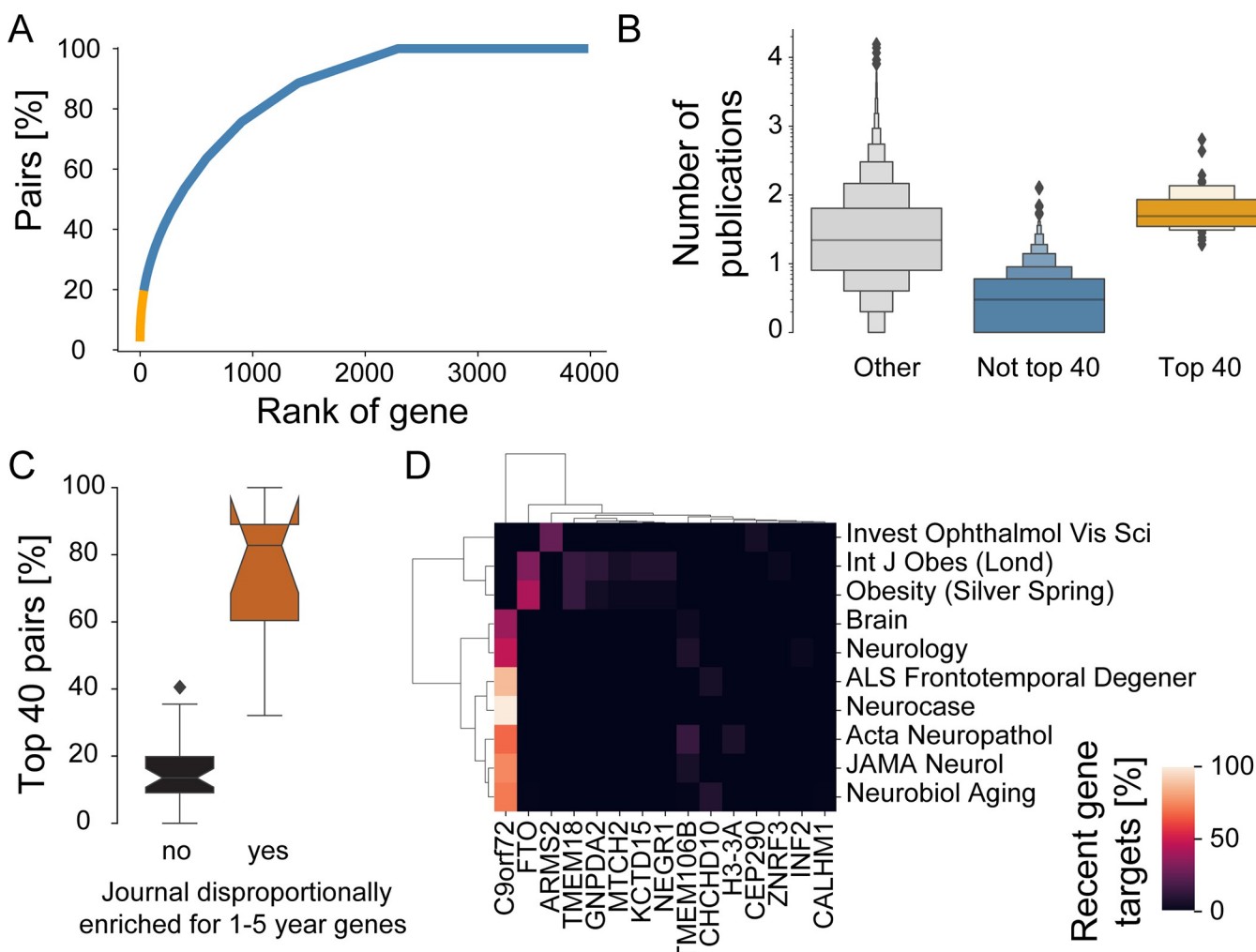

**Fig 6. Early-stage research focuses on a handful of genes and is driven by specific domains. (A)** Cumulative distribution of pairs of highlighted genes and 2010 to 2018 publications highlighting recent gene targets (first highlighted 1 to 5 years before). The x-axis ranks genes by number of publications highlighting them, with rank 0 corresponding to the gene highlighted in the most publications. The y-axis tallies the percentage of all pairs of highlighted genes and 2010 to 2018 publications highlighting recent gene targets with a lower or equal rank. We emphasize in orange the top 1% genes (40 genes), which together account for 21% of all pairs. Note that for simplicity of data representation, we allow for overlapping and shifting time windows. For example, a gene that has been first highlighted 5 years prior the start the indicated decade would will only be represented by a single year in our analysis. **(B)** Letter plot of all highlighted genes reporting for each gene the total number of publications until 2018 for genes that still were a recent target during the time 2010 to 2018 period, but were not ("Not top 40," blue) or were in the "top 40" ("Top 40," orange) and "other" genes (gray). Area of boxes indicates share of values, with heights of boxes following letter style dimensions (innermost boxes contain 25 to 75 percentiles of values and subsequent boxes 12.75 to 87.5 percentiles of values, etc.). Note that for 2010 to 2018, "other" is practically synonymous with genes first highlighted before 2005. Note further that some the genes of the "Not top 40" category may still accrue more publications until 2018, the final year of our analysis given the presence of shifting time windows. **(C)** Journals that enrich for top recent gene targets preferentially highlight top 40 genes. Percentage of pairs of publications and recent gene targets that fall onto the top 40 recently highlighted genes. Journals are considered to disproportionally enrich for recent gene targets if they enrich for recent gene targets at least twice as much as they enrich for novel gene targets (above dashed line in Fig 3C). **(D)** Different top 40 genes are highlighted in publications from journals dedicated to different fields. The heatmap plot shows the percentage of publications highlighting a given recently highlighted gene from among all publications in the given journal. Shown are journals that highlight recent gene targets and enrich for recent gene targets at least twice as much as they enrich for novel gene targets (above dashed line in Fig 3C). Genes and journals are ordered and grouped using Ward clustering. Combines data from MEDLINE, NCBI gene and taxonomy information, gene2pubmed, and PubTator. For data underlying the figure, see https://doi.org/10.21985/n2-b5bm-3b17.

gene implicated in age-related maculopathy. If we remove the 40 top ranked genes from our analysis, only 2 of these journals (*Brain* and *Obesity*) remain enriched for recent gene targets. In summary, we conclude that domain-specific research can contribute to early-stage research, but primarily does so through a handful of genes.

## Study limitations

We recognize that our investigation into early-stage research of human genes of the past 50 years cannot be separated from contemporary scientific research. We believe that we have identified a few specific ways in which contemporary scientific research shaped our findings. First, genes and their products are a primary focus of life scientists [25]. Second, our perspective toward genes follows the paradigms accepted by most US-based life scientists in 2021. Third, results in the scientific literature can be forgotten, and our analyses must be limited to those results that can be found in present day bibliometric resources. Fourth, the computational approaches to understand science likely do not capture all aspects of science and are biased toward entities that are easy to measure.

In addition to above general limitations, we also note that much of our investigation relies on individual research publications and note that life scientists frequently treat "research articles" synonymously with "research publications." Likewise, much of the scientometric literature focuses on research articles and measures success through peer recognition. Despite being prevalent, easily understandable, and a guide for some of our own analyses, such a view might be myopic as independent reevaluations of the data underlying research articles can be a valid and successful research strategy [58].

Moreover, we recognize that even on old gene targets, novel scientific findings can be made and that novel combinations of old scientific ideas can constitute innovation [35]. For instance, a novel connection between an old gene target and a previous unconnected disease could be just as impactful as an association between a disease and a novel gene target. Similarly, our study cannot provide direct guidance on what would be the most appropriate ratio between early-stage research and other modes of science.

## Discussion

Our analysis into the conditions surrounding early-stage research into human genes shows the need to be cautious when attempting to apply insights obtained from broad studies of innovation to narrower fields of science or research targets. Despite this need for caution, our study also demonstrates that it is possible to identify and quantify the contribution of individual research contexts toward innovation. An example of this is our finding that a discovery on a single gene (C9orf72) could be sufficient to lead to an extraordinary extent of innovation within neurobiology.

We find a set of journals and institutions that support research on novel gene targets and an even broader set that supports research into recent gene targets. As such, our findings are consistent, albeit in a more optimistic fashion, with the notion by de Grazia that scientific journals primarily serve individual communities of researchers and thereby develop "orthodox views . . . and that their receptivity to new ideas is variable" [45,46]. Specifically, we see that it is possible for some communities to become more receptive (such as genome research, and—currently—investigators of age-related neuropathies). Further, interdisciplinary journals, which avoid de Grazia's notion, support early-stage research to a larger extent. This demonstrates that journal policies can promote some forms of novelty. Last, the association between grants for postdoctoral researchers and early-stage research (S9 Fig) could hint at an opportunity for young scholars to build novel research communities around those domains of biomedical research that have a large potential to grow while avoiding direct competition with established senior investigators.

Our finding that grants for postdoctoral scholars promote innovation may align well with the approach of funding people rather than projects. This approach guides the Howard Hughes Medical Institute—one of the funding agencies we found to enrich for early-stage

research (S4A Fig)—and has been suggested to be favorable toward scientific innovation [27,59,60]. Irrespective of the specific policy to be promoted, early career scientists would appear to constitute an interesting target for funding-based policy interventions as early advantages in funding amplify over the course of the subsequent years [61].

Arguing against a straightforward dichotomy between people- and project-based funding, we found that project-based funding that is distributed through contracts also enriches for early-stage research to a similar extent as funding allocated through the Howard Hughes Medical Institute or grants of the NIH focused on postdoctoral scholars. In fact, most people-centered grants programs of the NIH enrich less for early-stage research on genes.

So, how could innovation in the research into human genes be promoted? Given our findings, it is tempting to suggest that greater investments in genome research would promote early-stage research. Yet, such a view is likely incomplete because innovation may be better viewed as a sequence of largely independent phases. Thus, the transitions between consecutive phases—rather than the innovativeness within individual phases—could determine the overall level of innovation. Plausibly, innovation in the research into human genes might be limited by barriers for the transitions between phases [11,14] (S6 Fig).

Supporting this view, the impact of genome-wide association studies on future research seems to be presently decreasing [62]. Efforts to promote the transition between early-stage research and subsequent research exist [10], but receive little financial support compared to other research [63]. Further, publicizing research opportunities might even discourage innovation by discouraging researchers from investigating these research opportunities because of the risk of being outcompeted by other researchers [64]. This study shows that successful transitions between early-stage research and subsequent research are not driven by any specific community (Fig 6). Rather, different scientific communities will respond to findings concerning novel gene targets. Whereas the small number of such successful transitions will preclude statistical analyses like those presented in this manuscript, we think that our identification of these successful transitions could allow for insight to be gained through focused historical and social studies.

The half-a-century view afforded by our analysis also demonstrates that even within a single scientific domain (research about human genes), specific findings about innovation can change over the course of a few decades. Therefore, continued efforts, and reanalyses of past studies with current data, might be needed to ensure that their insights are applicable to current day science. Opposing temporal and country-specific trends for a single domain of biology (such as cancer research) might even provide opportunity to identify processes underlying shifting policies. For instance, one may hypothesize that for some domains of biomedical research a focus on well-characterized genes could become desirable if meaningful societal impact stems from translational research, early diagnosis, or disease prevention [65].

Similarly, we do not dare to forecast whether genome-wide association studies will continue to be prominent among early-stage research. We envision at least 2 alternative, complementary directions, which stem from technology that currently gains popularity independent of early-stage research, but would appear to be particularly well positioned to prompt the investigation of novel gene targets. First, genes with expression patterns that are tissue or cell type specific remain little characterized [11]. Promisingly, single-cell transcriptomic approaches would seem well positioned to provide insight into the function of these genes and identify cell populations where those genes could be of physiological relevance. Second, current insight into physiology appears to primarily reflect top-down approaches (e.g., where a phenotype of interest is known but underlying factors might not be known), whereas bottom-up approaches seem risky as there could be myriad possibilities on what a given gene might do, and many of these possibilities may not be very interesting. However, computational predictions of protein

structures are already extending toward multimers and complexes [66,67], and therefore, it might be within sight a time when the binding potentials of all pairs of proteins will be known. Genome-wide structural bottom-up approaches would seem ideally positioned to answer whether uncharacterized and novel gene targets are part of known complexes or hint at novel aspects of biology.

Our analysis also foreshadows that innovative research into human genes will soon no longer be appraisable through the novelty of its gene targets. Ultimately, we are reminded of de Solla Price's observation that future trajectories of scientific disciplines seem uncertain once these fields will have slowed down their growth [15]. He noted that these trajectories could range from alternating short phases of growth and recession to the disappearance of the scientific field or their stagnation. Consequently, research into genes and their molecular products might be headed toward turbulent times.

Aside from some important but rare exceptions [52,53,57] (Fig 6), the wealth of unbiased data that has already been accumulated through genome biology currently stands in stark contrast to the genes that are presently chosen for consistent—or even insistent—investigation by life scientists [11,14,24]. We thus anticipate that domain-specific scientists—but also the funding agencies, research organizations, and foundations supporting their work—will all increasingly need to revisit the research patterns within their domains to better justify their research to the public. We hope that an improved and more specific understanding of early-stage research will help to create ideas and hypotheses on how to counter unintended patterns of scientific inquiry.

## Methods

### Literature parsing

Publication dates, journal names, titles, abstracts, author-provided keywords, MeSH terms, number of authors, funders, and publication types were obtained from MEDLINE through the PubMed baseline files downloaded on October 21, 2020 from https://www.nlm.nih.gov/databases/download/pubmed_medline.html and subsequently parsed through PubMed Parser 0.2.2 [68].

Out of all 30,419,065 publications that we retrieved, we excluded 18 publications because the PubMed ID was duplicated. We further removed 1,334,568 publications published after 2018 before filtering our data further as described below to identify the publications with human protein-coding genes in their title or abstract.

Genes in titles or abstracts were obtained through PubTator [69]. Specifically, we downloaded https://ftp.ncbi.nlm.nih.gov/pub/lu/PubTatorCentral on October 22, 2020 and used the therein contained offsets of recognized entities (such as genes) to locate whether they fall into the title or abstract. We excluded ambiguously mapped entities indicated by an ";" operator in PubTator.

To identify research publications, we excluded publications annotated with any of the following publication type in MEDLINE: D016454:Review, D016422:Letter, D016420:Comment, D016421:Editorial, D016456:Historical Article, D016433:News, D019215:Biography, D016425: Published Erratum, D054711:Introductory Journal Article, D019477:Portrait, D017203:Interview, D016424:Overall, D017065:Practice Guideline, D059040:Video-Audio Media, D018431: Newspaper Article, D016417:Bibliography, D016440:Retraction of Publication, D016441: Retracted Publication, D019531:Lecture, D016435:Directory, D016419:Classical Article, D019484:Address, D029282:Patient Education Handout, D062210:Personal Narrative, D020493:Autobiography, D020485:Legislation, D016221:Festschrift, D016438:Duplicate Publication, D057405:Webcast, D064886:Dataset, D016437:Dictionary, D000075742:Expression of

Concern, D016453:Periodical Index, D054710:Interactive Tutorial, D016426:Scientific Integrity Review, D020470:Collected Works, D057405:Webcasts.

### Literature database construction

To trace early-stage research into human protein-coding genes, we will follow genes that are referred to by their names in the titles and abstracts of scientific publications. This choice is motivated by pragmatic and scientific reasons. First, titles and abstracts are broadly available and accessible to computational approaches. Second, scientific authors take great care while preparing those section of scientific publications [70], whereas readers have been advised to quickly read these sections prior to committing to reading the entire scientific publication [71]. We therefore surmise that the genes mentioned within the title or abstract have been highlighted by the authors and are considered by them to be related to the most important research results reported in the scientific publication.

We acknowledge that past scientific literature can become detached from future literature [72] and that throughout the past 50 years, the concept of genes has itself remained subjected to variable definitions that extended into efforts of distinct groups of scientists to claim how and when genes should be named [73–77]. While we believe that those matters warrant continued historical studies, we will, for the present manuscript, sidestep them in a formalized manner by restricting our inquiry to present day indices of scientific literature and genes, which the National Library of Medicine and the National Center for Biotechnology Information of the US offer to life scientists as references for the biomedical literature and genes. Consequently, we would like to caution readers of this manuscript that our investigation into the conditions surrounding early-stage research into human protein-coding during the past 50 years will adopt a selective and gene-centric perspective that is possibly biased toward the perspective of American life scientists of 2021 (see "Study limitations" section).

We first use gene2pubmed [78] to identify scientific publications on human genes. Next we use PubTator [69] to identify genes within the title or abstract of those publications. We end our analysis in 2018 as more recent publications appear to not have yet been fully indexed by gene2pubmed.

Overall, we consider 485,800 PubMed scientific publications that yield 1,265,150 unique genes–article pairs. A total of 61% of all publications only highlight 1 or 2 genes, and no publication highlights more than 56 genes (S1A Fig). We manually reviewed 100 randomly sampled scientific publications for correctness of gene–publication pairs and find a false-positive rate of 5%. We additionally manually reviewed 50 randomly sampled scientific publications that were annotated by gene2pubmed for human genes but for which there were no highlighted genes according to PubTator and found 19 true false negatives. Notably, for the latter analysis, we only consider publications of gene2pubmed linked to protein-coding genes, the focus of our investigation—thereby slightly reducing the overall number of publications (to 473,477). Extrapolating the 19 true false negatives to the 98,803 scientific publications of the 572,280 publications of gene2pubmed showing no overlap between gene2pubmed and PubTator, we estimate an upper boundary for the false-negative rate in identifying highlighted genes to be below 11%. We conclude that our approach captures most human genes highlighted in the titles or abstracts of indexed research publications and provides for high-quality data for this study.

To investigate the extent to which our investigation of past research is constrained by present day life sciences knowledge, we randomly select for each decade 10 genes that had not been highlighted prior to that decade and performed a manual literature review for those genes (S1B Fig). Prior to 2000, we encounter a large degree of ambiguity in the names of genes.

We classify a naming situation as "misty" if we find that heritable traits, symptoms, enzymatic activities, mutants, biochemical affinities, clone identifiers, domains, cosegregation, partial sequence information, or loci relating to these genes have been named in titles or abstracts before, but that those related entities have—in contrast to such entities relating to other genes —neither become a preserved or accepted namesake or synonym of the respective genes. Additionally, we observe 4 cases where our approach "missed" a preceding highlighting of a gene. One case is due to a misattribution of the gene to another species by gene2pubmed, and 2 other cases stem from PubTator not indexing the publications, and the final case stems from PubTator not having detected the gene. Reassuringly, in all these cases, the research publications identified by our pipeline still describe early-stage research that either established or manifested the existence of these genes or established them to exist as entities that are separate from other genes.

Further, we encounter 4 cases that we classify as "missed by name" where multiple genes shared a name or the name of one gene became part of the name of another gene. Also, in these cases, at least 1 of the highlighted genes may be classified as being in early stages of investigation, although some of these cases retain further ambiguity. For instance, PubTator highlights "transferrin" in a research publication about "transferrin receptor." Transferrin had already been known at the time, but its gene had not yet been cloned.

### Definition of genes

We considered NCBI gene info (https://ftp.ncbi.nlm.nih.gov/gene/DATA/gene_info.gz) downloaded on June 21, 2020, homologene build 68 (https://ftp.ncbi.nlm.nih.gov/pub/HomoloGene/build68), and NCBI taxonomy (https://ftp.ncbi.nlm.nih.gov/pub/taxonomy) downloaded on June 30, 2020.

### Loss-of-function intolerance

We considered pLI scores above 0.9 within the intolerances published by Karczewski and colleagues [32].

### Phenotype databases

For human phenotypes, we used the NHGRI-EBI catalog of genome-wide association studies [34] (https://www.ebi.ac.uk/gwas) downloaded on May 9, 2021.

For mouse phenotypes, we used IMPC release 13.0 [33] (https://www.mousephenotype.org/data/previous-releases/13.0).

### Statistical analysis

Analysis was performed in sciPy v.1.4 [79].

Specific metrics are given in figure legends and https://doi.org/10.21985/n2-b5bm-3b17, which also details the specific way we obtained confidence interval to indicate the robustness of our findings.

We performed no further sensitivity analysis and note that conveyed confidence intervals depend upon the inherent variability within a group and the number of publications in that group. As the number of publications has been increasing over recent decades, our approach should be more sensitive for more recent decades.

### Data visualization

Data was visualized through seaborn 0.11 [80].

## Citation analysis

For relative and relational citation data, we used iCite Version 10 [81]. We defined studies as belonging to a clinical trial if their publication type on PubMed would contain "clinical trial" in a case-insensitive manner.

## Identification of NIH-supported research organizations

We downloaded public funding information, including funding activity codes, and supported research organizations and the mapping between publications and research grants through NIH's https://exporter.nih.gov on November 3, 2020.

## Definition of genome biology

We considered a publication to be genome biology if title, abstract, author-submitted keywords, or MeSH terms contained one of the following phrases in a case-insensitive manner. Their selection follows a manually supervised identification of terms that are enriched to co-occur with genomics and/or which we spotted in review articles on genomics.

chip seq, chip sequencing, chip-chip, chip-seq, chip-sequencing, chromatin immunoprecipitation followed by sequencing, clinicogenomic, clinicogenomics, clip seq, clip seq, clip sequencing, clip sequencing, clip-seq, clip-seq, clip-sequencing, clip-sequencing, epigenome, epigenomes, epigenomic, epigenomics, eqtl, eqtls, exome, expression quantitative trait loci, ewas, genome, genome-scale, genome-wide, genomes, genomic, genomics, glycome, glycomes, glycomics, glycoproteome, glycoproteomes, glycoproteomic, glycoproteomics, gwas, high-throughput nucleotide sequencing, hits seq, hits sequencing, hits-seq, hits-sequencing, in situ proximity ligation, in-situ proximity ligation, interactome, interactomes, interactomic, interactomics, metabolome, metabolomes, metabolomic, metabolomics, metagenome, metagenomes, metagenomics, microarray, microarrays, multi-ome, multi-omes, multi-omic, multi-omics, multiome, multiomes, multiomic, multiomics, next generation sequencing, next generation-sequencing, next-generation sequencing, next-generation-sequencing, ngs, nutrigenome, nutrigenomes, nutrigenomic, nutrigenomics, oligonucleotide array sequence analysis, omics, onco-genome, onco-genomes, onco-genomics, onco-genomics, onco-proteogenome, onco-proteogenomes, onco-proteogenomic, onco-proteogenomics, oncogenome, oncogenomes, oncogenomics, oncogenomics, oncoproteogenome, oncoproteogenomes, oncoproteogenomic, oncoproteogenomics, par-clip, pharmacogenome, pharmacogenomes, pharmacogenomic, pharmacogenomics, phenome, phenomes, phenomic, phenomics, phosphoproteome, phosphoproteomes, phosphoproteomic, phosphorpoteomics, protein array, protein array analysis, protein interaction map, protein interaction mapping, protein interaction maps, protein interaction network, protein interaction networks, protein–protein interaction map, protein–protein interaction mapping, protein–protein interaction maps, protein–protein interaction network, protein–protein interaction networks, proteogenome, proteogenomes, proteogenomic, proteogenomics, proteome, proteomes, proteomic, proteomics, radiogenome, radiogenomes, radiogenomic, radiogenomics, rna seq, rna sequencing, rna-seq, rna-sequencing, rnaseq, toxicogenome, toxicogenomes, toxicogenomic, toxicogenomics, transcriptome, transcriptomes, transcriptomic, transcriptomics, wes, wgs, whole-exome, whole-genome.

## Definition of genome-wide association study

We considered a publication to be a genome-wide association study if it carried the MeSH term D055106 (which corresponds to genome-wide association studies) or if the publication

appeared in the NHGRI-EBI catalog of genome-wide association studies [34] or if title, abstract, or author-submitted keywords contained one of the following phrases in a case-insensitive manner:

gwas, genome wide association study, genome-wide association study, GWA study, gwa studies, whole genome association study, genome wide association scan, genome wide association analysis, genome-wide association scan, genome-wide association analysis, twas, transcriptome wide association study, transcriptome-wide association study, twa study, twa studies, whole transcriptome association study, transcriptome wide association scan, transcriptome wide association analysis, transcriptome-wide association scan, transcriptome-wide association analysis.

## Supporting information

**S1 PRISMA Checklist.**
(DOCX)

**S1 Fig. Quantifying some of the limitations of the approach pursued in this study. (A)** Survival analysis share of publications highlighting a gene in the title or abstract that highlight at least the indicated number of genes. **(B)** Estimation of impact of changes in gene nomenclature and other issues in determining whether a given gene is highlighted in a publication or not. For each decade, we randomly selected 10 genes that our automated approach did not identify as being highlighted by name in the title or abstract prior to the year of the oldest automatically identified publication. We mark in yellow "clear cases," where past nomenclature aligns with today's nomenclature and for which we could not find any preceding highlighting. We mark in gray "foggy cases," where we found that entities related to those genes (e.g., symptoms, enzymatic activities,. . .) had been highlighted before, but—in contrast to some other genes—are not currently accepted as namesakes or synonyms of these genes by NCBI Gene. We mark in pink "missed by name cases," where the gene highlighted in a publication was misattributed because the 2 genes either swapped names later or the name of one gene became part of the name of the other gene. Finally, we mark in black "missed cases," where an earlier highlighting of the genes was missed by our automated computational approach. Combines data from MEDLINE, NCBI gene and taxonomy information, gene2pubmed, and PubTator. For data underlying the figure, see https://doi.org/10.21985/n2-b5bm-3b17.
(PDF)

**S2 Fig. Percentage of annually highlighted genes. (A)** Percentage of protein-coding genes highlighted in a given year in at least 1 publication. **(B)** Percentage of protein-coding genes highlighted in a given year in at least 1 publication that highlights a single gene. **(C)** Share of human loss-of-function intolerant protein-coding genes which have been highlighted (mentioned by name in title or abstract) until the indicated year (solid) or mentioned by name in the title (dotted). Combines data from MEDLINE, NCBI gene and taxonomy information, ene2pubmed, PubTator, and Karczewski and colleagues. [32]. For data underlying the figure, see https://doi.org/10.21985/n2-b5bm-3b17.
(PDF)

**S3 Fig. The rate at which genes are highlighted in publications is highly correlated across time and with year of first highlight. (A)** Comparison of the number of articles highlighting a particular gene for 2009 to 2018 publications versus for 1980 to 2008 publications. **(B)** Comparison of the number of articles highlighting a particular gene for 2009 to 2018 publications versus year gene was first highlighted in a publication. Rho ($\rho$) indicates Spearman correlation coefficient. Combines data from MEDLINE, NCBI gene and taxonomy information,

gene2pubmed, and PubTator. For data underlying the figure, see https://doi.org/10.21985/n2-b5bm-3b17.
(PDF)

**S4 Fig. Loss-of-function intolerant genes have become underrepresented in the literature in the late 1990s. (A)** Enrichment ($\log_2$ of ratio) of publications highlighting at least 1 loss-of-function intolerant gene (solid lines) aggregated by gene age and publication year (obtained as an average over a 3-year moving window). Loss-of-function intolerance was inferred from human polymorphisms. The solid lines show enrichment for loss-of-function intolerant genes, whereas for comparison the dashed lines show enrichment for genes that are not loss-of-function intolerant. Gene ages at publication are grouped identically in both panels A and B and as shown in legend of panel A. Error bars show 95% confidence intervals inferred by bootstrap. **(B)** As (A) but with loss-of-function intolerance inferred from the occurrence of non–wild-type phenotypes in systematic murine mutagenesis experiments. **(C)** Enrichment ($\log_2$ of ratio) of annual pairs of highlighted genes and research publications relative to the number of genes and the number of different phenotypes in a systematic murine mutagenesis experiment. **(D)** Enrichment ($\log_2$ of ratio) of annual pairs of highlighted genes and research publications relative to the number of genes and the number of different traits or disease in the NHGRI-EBI catalog of genome-wide association studies [34]. Note that C and D show 95% confidence intervals inferred by bootstrap of the genes but that bootstrap only has minimal effect on the enrichment. Combines data from MEDLINE, NCBI gene and taxonomy information, gene2-pubmed, PubTator, Karczewski and colleagues [32], IMPC [33], and NHGRI-EBI GWAS catalog. For data underlying the figure, see https://doi.org/10.21985/n2-b5bm-3b17.
(PDF)

**S5 Fig. Publications reporting early-stage research on human genes differ in the number of authors and the number of genes highlighted in the publication from publications highlighting older genes. (A)** Number of authors for publications highlighting at least 1 gene aggregated by gene age and publication decade. **(B)** Number of highlighted genes for publications highlighting at least 1 gene aggregated by gene age and publication decade. Notches in box plots indicate 95% confidence interval of the median. Gene ages at publication are grouped identically in both panels and as shown in legend of panel A. Combines data from MEDLINE, NCBI gene and taxonomy information, gene2pubmed, and PubTator. For data underlying the figure, see https://doi.org/10.21985/n2-b5bm-3b17.
(PDF)

**S6 Fig. Publications reporting early-stage research are distinctively received. (A)** Percentile of citations to publications highlighting at least 1 gene aggregated by age of highlighted gene and by decade. Notches indicate 95% confidence intervals of the median. **(B)** Citation variability as determined by the width of the interquartile range of the citations, normalized by percentiles of the year of the publication (with width being span between 25 and 75 percentiles). Shaded area indicates 95% confidence intervals inferred by bootstrap. Year indicates center of a 3-year sliding window used for analysis. **(C)** Years until first citation by a clinical trial to a publication highlighting at least 1 gene aggregated by age of highlighted gene and by decade. **(D)** Percentile of disruption index of a publication highlighting at least 1 gene aggregated by age of highlighted gene and by decade. Brown indicates publications with at least 1 gene that has not been highlighted in any preceding year. Dark orange indicates publications with at least 1 gene that was first highlighted during the 5 preceding years. Light orange indicates publications that only highlight genes that have been first highlighted 6 or more years earlier. Combines data from MEDLINE, NCBI gene and taxonomy information, gene2pubmed,

PubTator and iCite. For data underlying the figure, see https://doi.org/10.21985/n2-b5bm-3b17.
(PDF)

**S7 Fig. Identification of journals that associate with early-stage research by decade of publication.** Plots within the left column focus on publications that highlight at least 1 new gene that had not been highlighted in any preceding year (0-year genes), whereas plots within the right column focus on publications that highlight at least 1 gene that was recently highlighted for the first time (1- to 5-year genes). In each plot, we compare the number of publications in a given journal highlighting at least 1 new or recent gene target against the total number of other publications in the same journal. Each circle represents an individual journal that published at least 1 study highlighting at least 1 gene. Red (cyan) circles indicate journals significantly enriched (depleted) for respective early-stage research. Combines data from MEDLINE, NCBI gene and taxonomy information, gene2pubmed, and PubTator. For data underlying the figure, see https://doi.org/10.21985/n2-b5bm-3b17.
(PDF)

**S8 Fig. Identification of funding agencies that associate with early-stage research by decade of publication.** Plots within the left column focus on publications that highlight at least 1 new gene that had not been highlighted in any preceding year (0-year genes), whereas plots within the right column focus on publications that highlight at least 1 gene that was recently highlighted for the first time (1- to 5-year genes). In each plot, we compare the number of publications acknowledging funding from a given agency highlighting at least 1 new or recent gene against the total number of other publications acknowledging funding from the same agency. Each circle represents a funding agency acknowledged in a least 1 publication that highlighting at least 1 gene. Red (cyan) circles indicate funding agency significantly enriched (depleted) for respective early-stage research. Combines data from MEDLINE, NCBI gene and taxonomy information, gene2pubmed, PubTator, and ExPORTER.
(PDF)

**S9 Fig. Identification of activity codes from NIH funding that associate with early-stage research by decade of publication.** Plots within the left column focus on publications that highlight at least 1 new gene that had not been highlighted in any preceding year (0-year genes), whereas plots within the right column focus on publications that highlight at least 1 gene that was recently highlighted for the first time (1- to 5-year genes). In each plot, we compare the number of publications acknowledging funding from a given activity code highlighting at least 1 new or recent gene against the total number of other publications acknowledging funding from the same activity. Each circle represents an activity code for NIH funding acknowledged in a least 1 publication that highlighting at least 1 gene. Red (cyan) circles indicate activity codes significantly enriched (depleted) for respective early-stage research. Combines data from MEDLINE, NCBI gene and taxonomy information, gene2pubmed, and PubTator. For data underlying the figure, see https://doi.org/10.21985/n2-b5bm-3b17. NIH, National Institutes of Health.
(PDF)

**S10 Fig. Inspection of activity codes enriching for early-stage research. (A)** Comparison of fold enrichment ($\log_2$ of ratio) for activity codes from NIH funding that are significantly enriched or depleted for new gene targets or recent gene targets. Note that for any given activity code, the enrichment may not be statistically significant for both axes. **(B)** Box plot showing share of grants—measured through fiscal years (as grant durations can vary among activity codes)—that go toward enriched activity codes between 2010 and 2018 (salmon box of panel

A) for NIH institutes that do not (no) or do (yes) enrich for early-stage research. *p*-Value is obtained by 2-sided Mann–Whitney U test. Combines data from MEDLINE, NCBI gene and taxonomy information, gene2pubmed, PubTator, and ExPORTER. For data underlying the figure, see https://doi.org/10.21985/n2-b5bm-3b17. NIH, National Institutes of Health. (PDF)

**S11 Fig. Identification of author institutional affiliations that associate with early-stage research by decade of publication.** Plots within the left column focus on publications that highlight at least 1 new gene that had not been highlighted in any preceding year (0-year genes), whereas plots within the right column focus on publications that highlight at least 1 gene that was recently highlighted for the first time (1- to 5-year genes). In each plot, we compare the number of publications with at least 1 author affiliated with a given institution highlighting at least 1 new or recent gene against the total number of other publications with at least 1 author affiliated with the same institution. Each circle represents an institution associated with at least 1 publication that highlighting at least 1 gene. Red (cyan) circles indicate institutions significantly enriched (depleted) for respective early-stage research. Combines data from MEDLINE, NCBI gene and taxonomy information, gene2pubmed, PubTator, and ExPORTER. For data underlying the figure, see https://doi.org/10.21985/n2-b5bm-3b17. (PDF)

**S12 Fig. Percentage of publications that are categorized as genome biology highlighting at least 1 gene aggregated by age of highlighted gene and by decade.** Shaded area indicates 95% confidence intervals inferred by bootstrap. Combines data from MEDLINE, NCBI gene and taxonomy information, gene2pubmed, and PubTator. For data underlying the figure, see https://doi.org/10.21985/n2-b5bm-3b17. (PDF)

**S13 Fig. Early-stage research tends to be produced by larger scientific teams and highlights more genes. (A)** Mean number of authors for publications categorized as genome biology highlighting at least 1 gene aggregated by age of highlighted gene and by decade. **(B)** Mean number of authors for publications not categorized as genome biology highlighting at least 1 gene aggregated by age of highlighted gene and by decade. **(C)** Box plot of number of authors for publications categorized as genome biology highlighting at least 1 gene aggregated by age of highlighted gene and by decade. **(D)** Box plot of number of authors for publications not categorized as genome biology highlighting at least 1 gene aggregated by age of highlighted gene and by decade. **(E)** Mean number of highlighted genes for publications categorized as genome biology highlighting at least 1 gene aggregated by age of highlighted gene and by decade. **(F)** Mean number of highlighted genes for publications not categorized as genome biology highlighting at least 1 gene aggregated by age of highlighted gene and by decade. **(G)** Box plot of number of highlighted genes for publications categorized as genome biology highlighting at least 1 gene aggregated by age of highlighted gene and by decade. **(H)** Box plot of number of highlighted genes for publications not categorized as genome biology highlighting at least 1 gene aggregated by age of highlighted gene and by decade. Error bars show 95% confidence intervals inferred by bootstrap. Gene ages at publication are grouped identically in all panels and as shown in legend of panel A. Notches in box plots indicate 95% confidence intervals of the median. n/a indicates non applicability due to absence of genome biology publications in 1970 to 1979. Combines data from MEDLINE, NCBI gene and taxonomy information, gene2pubmed, and PubTator. For data underlying the figure, see https://doi.org/10.21985/n2-b5bm-3b17. (PDF)

**S14 Fig. Publications reporting early-stage research is more likely to be highly cited. (A)** Percentage of highly cited publications categorized in genome biology (among top 5% of indicated year) aggregated by decade and age of highlighted gene. **(B)** Percentage of highly cited publications not categorized in genome biology (among top 5% of publication year) aggregated by decade and age of highlighted gene. Dashed line shows 5% baseline. Gene ages at publication are grouped identically in all panels and as shown in legend of panel A. n/a indicates non applicability due to absence of genome biology publications in 1970 to 1979. Combines data from MEDLINE, NCBI gene and taxonomy information, gene2pubmed, PubTator, and iCite. For data underlying the figure, see https://doi.org/10.21985/n2-b5bm-3b17. (PDF)

**S15 Fig. Within genome research, genome-wide association studies tend to produce early-stage research with larger scientific teams and to highlight more genes. (A)** Mean number of authors for publications categorized as genome-wide association study highlighting at least 1 gene aggregated by age of highlighted gene and by decade. **(B)** Mean number of authors for publications categorized as not being a genome-wide association study highlighting at least 1 gene aggregated by age of highlighted gene and by decade. **(C)** Box plot of number of authors for publications categorized as genome-wide association study highlighting at least 1 gene aggregated by age of highlighted gene and by decade. **(D)** Box plot of number of authors for publications categorized as not being a genome-wide association study highlighting at least 1 gene aggregated by age of highlighted gene and by decade. **(E)** Mean number of highlighted genes for publications categorized as genome-wide association study highlighting at least 1 gene aggregated by age of highlighted gene and by decade. **(F)** Mean number of highlighted genes for publications categorized as not being a genome-wide association study highlighting at least 1 gene aggregated by age of highlighted gene and by decade. **(G)** Box plot of number of highlighted genes for publications categorized as genome biology and genome-wide association study highlighting at least 1 gene aggregated by age of highlighted gene and by decade. **(H)** Box plot of number of highlighted genes for publications categorized as not being a genome-wide association study highlighting at least 1 gene aggregated by age of highlighted gene and by decade. Error bars show 95% confidence intervals inferred by bootstrap. Gene ages at publication are grouped identically in all panels and as shown in legend of panel A. Notches in box plots indicate 95% confidence intervals of the median. n/a indicates non applicability due to absence of genome biology publications in 1970 to 1979 and genome-wide association studies in 1970 to 1999. Combines data from MEDLINE, NCBI gene and taxonomy information, gene2pubmed, and PubTator. For data underlying the figure, see https://doi.org/10.21985/n2-b5bm-3b17. (PDF)

**S16 Fig. Within genome research, genome-wide association studies reporting early-stage research are more likely to be highly cited. (A)** Percentage of highly cited publications (among top 5% of indicated year) for publications categorized as genome-wide association study aggregated by decade and age of highlighted gene. **(B)** Percentage of highly cited (among top 5% of publication year) publications for publications not categorized as not being a genome-wide association study aggregated by decade and age of highlighted gene. Dashed line shows 5% baseline. Gene ages at publication are grouped identically in all panels and as shown in legend of panel A. n/a indicates non applicability due to absence of genome biology publications in 1970 to 1979 and genome-wide association studies in 1970 to 1999. Combines data from MEDLINE, NCBI gene and taxonomy information, gene2pubmed, PubTator, and iCite. For data underlying the figure, see https://doi.org/10.21985/n2-b5bm-3b17. (PDF)

**S17 Fig. The top 1% genes of every decade account for a high percentage of publications reporting early-stage research.** Note that the number of genes in the top 1% increases over time because so does the total number of gene highlighted in each period. (Left) Cumulative distribution of pairs of highlighted genes and publications in considered period highlighting recent gene targets (first highlighted 1 to 5 years before). The x-axis ranks genes by number of publications highlighting them, with rank 0 corresponding to the gene highlighted in the most publications within a given decade. The y-axis tallies the percentage of all pairs of highlighted genes in publications in considered period highlighting recent gene targets with a lower or equal rank. We emphasize in orange the top 1% genes which together account for over 15% of all pairs. (Right) Letter plots of all highlighted genes reporting for each gene the total number of publications until 2018 for genes that still were a recent target during the time period of the respective left panel with top 1% genes in orange and other 99% gene in orange. For comparison, all other genes (which are no recent gene target during time period) are shown in gray. Area of boxes indicates share of values, with heights of boxes following letter style dimensions —innermost boxes contain 25 to 75 percentiles of values and subsequent boxes 12.75 to 87.5 percentiles of values, etc. Combines data from MEDLINE, NCBI gene and taxonomy information, gene2pubmed, and PubTator. For data underlying the figure, see https://doi.org/10.21985/n2-b5bm-3b17.
(PDF)

**S1 PRISMA Flow Diagram.**
(DOCX)

## Acknowledgments

We thank all members of the Amaral lab for feedback. Additionally, we thank Rick Morimoto and Christopher Donohue for feedback on an early draft.

## Author Contributions

**Conceptualization:** Thomas Stoeger, Luís A. Nunes Amaral.

**Data curation:** Thomas Stoeger.

**Formal analysis:** Thomas Stoeger.

**Funding acquisition:** Thomas Stoeger, Luís A. Nunes Amaral.

**Investigation:** Thomas Stoeger, Luís A. Nunes Amaral.

**Methodology:** Thomas Stoeger, Luís A. Nunes Amaral.

**Project administration:** Luís A. Nunes Amaral.

**Resources:** Thomas Stoeger.

**Supervision:** Luís A. Nunes Amaral.

**Visualization:** Thomas Stoeger, Luís A. Nunes Amaral.

**Writing – original draft:** Thomas Stoeger, Luís A. Nunes Amaral.

**Writing – review & editing:** Thomas Stoeger, Luís A. Nunes Amaral.

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
