## [Editor Report · Decision Letter 0]

1 Oct 2021

Dear Thomas, 

Thank you for submitting your manuscript entitled "Early-stage research into human genes differs in team size, citation impact, funding mechanisms, publication outlet, and scientific approach" for consideration as a Meta-Research Article by PLOS Biology.

Your manuscript has now been evaluated by the PLOS Biology editorial staff, as well as by an academic editor with relevant expertise, and I'm writing to let you know that we would like to send your submission out for external peer review.

Once your full submission is complete, your paper will undergo a series of checks in preparation for peer review. Once your manuscript has passed the checks it will be sent out for review. 

If your manuscript has been previously reviewed at another journal, PLOS Biology is willing to work with those reviews in order to avoid re-starting the process. Submission of the previous reviews is entirely optional and our ability to use them effectively will depend on the willingness of the previous journal to confirm the content of the reports and share the reviewer identities. Please note that we reserve the right to invite additional reviewers if we consider that additional/independent reviewers are needed, although we aim to avoid this as far as possible. In our experience, working with previous reviews does save time. 

If you would like to send your previous reviewer reports to us, please specify this in the cover letter, mentioning the name of the previous journal and the manuscript ID the study was given, and include a point-by-point response to reviewers that details how you have or plan to address the reviewers' concerns. Please contact me at the email that can be found below my signature if you have questions. 

Please re-submit your manuscript within two working days, i.e. by Oct 05 2021 11:59PM.

Best wishes,

Roli

Roland Roberts

Senior Editor

PLOS Biology

rroberts@plos.org

---

## [Decision Letter · Decision Letter 1]

25 Nov 2021

Dear Thomas,

Thank you for submitting your revised Meta-Research Article entitled "Early-stage research into human genes differs in team size, citation impact, funding mechanisms, publication outlet, and scientific approach" for publication in PLOS Biology. I have now obtained advice from two of the original reviewers and have discussed their comments with the Academic Editor. 

Based on the reviews, we will probably accept this manuscript for publication, provided you satisfactorily address the remaining points raised by the reviewers. Please also make sure to address the following data and other policy-related requests.

IMPORTANT:

a) We find your current Title rather obscure ("differs from what?" is the question that springs to mind) and the list of properties is cumbersome. We suggest the following options (or you may also have some ideas): "Early-stage research into human genes differs from subsequent work in team size, citation impact, funding mechanisms, publication outlet, and scientific approach" (still long and cumbersome), or "The characteristics of early-stage research into human genes are substantially different from subsequent research" or even "Field-wide analysis shows that early-stage research into humans genes has substantially different characteristics from subsequent research." The middle one of these is probably our preferred option.

b) If you choose a Title that lacks the list of properties, then you could include these (i.e. team size, citation impact, funding mechanisms, publication outlet, and scientific approach) by expanding the Abstract.

c) Please attend to all of the remaining requests from reviewer #1. Reviewer #3 has no further requests.

d) Please address my Data Policy requests below; specifically, while we recognise that the raw data fall under our third-party data exemption, we need you to supply the numerical values that directly underlie Figs 1ABC, 2ABCD, 3ABC, 4ABCD, 5ABCDE, 6ABCD, S1AB, S2ABC, S3AB, S4AB, S5AB, S6ABCD, S7, S8, S9, S10AB, S11, S12, S13ABCDEF, S14AB, S15. Please cite the location of the data clearly in each relevant Fig legend.

e) Because this is a meta-analysis, we need you to include a completed PRISMA checklist and flow diagram (as supplementary files). See the policy here: https://journals.plos.org/plosbiology/s/best-practices-in-research-reporting#loc-reporting-guidelines-for-specific-study-types

We expect to receive your revised manuscript within two weeks. 

*Published Peer Review History*

*Early Version*

Sincerely,

Roli

Senior Editor,

rroberts@plos.org,

PLOS Biology

DATA POLICY:

Note that we do not require all raw data; indeed we recognise that most of these are third-party, and therefore fall under the corresponding exemption clause of PLOS' data policy, as long as they can be accessed in the same manner as you did. Rather, we ask that all numerical values that directly underlie the figures and results of your paper be made available in one of the following forms:

Regardless of the method selected, please ensure that you provide the individual numerical values that underlie the summary data displayed in the following figure panels as they are essential for readers to assess your analysis and to reproduce it: Figs 1ABC, 2ABCD, 3ABC, 4ABCD, 5ABCDE, 6ABCD, S1AB, S2ABC, S3AB, S4AB, S5AB, S6ABCD, S7, S8, S9, S10AB, S11, S12, S13ABCDEF, S14AB, S15. NOTE: the numerical data provided should include all replicates AND the way in which the plotted mean and errors were derived (it should not present only the mean/average values).

DATA NOT SHOWN?

REVIEWERS' COMMENTS:

Reviewer #1:

In this paper, Stoeger and Amaral analyze the contexts that have historically enabled early-stage research into human genes to better understand how such research can be sustained and revitalized in the contemporary research ecosystem. I have a few comments/questions about the overall framing of the manuscript and how much these results might be driven by GWAS publications in the last ~15 years, but overall I think it is a thorough and interesting contribution to the metaresearch literature, and should spark some healthy debate about balancing depth and breadth in genetics/genomics research and the surrounding incentive structures.

Major comments:

1. The paragraph in lines 38-49 is extremely unclear to me. The authors state that the narrow focus on well-studied genes raises "significant concerns" about knowledge generation. The examples the authors cite to motivate this claim, however, seem quite anecdotal and/or poorly communicated. First, the sentence in lines 39-41 (with reference to Ioannidis, 2005) seems to be insinuating that results reported for well-studied genes are increasingly enriched for false positives, relative to the results reported for understudied genes. Is there any evidence that the false positive *rate* correlates with a gene's popularity in the literature? This would indeed be troubling if true, but it seems quite speculative. Also, the authors suggest that incorrect research results are selectively accumulating among already well-studied genes, but the claim that this is a "significant concern" overlooks a lot of important context--perhaps such incorrect results are primarily appearing in low-impact or predatory journals (e.g., Genet Mol Res, as described in lines 270-272) or are enriched in publications from a small subset of authors with poor data or incorrect assumptions. Regardless, this claim feels too strong without some empirical contrast showing that the false positive rate is higher for well-studied genes and/or those false positives are qualitatively more detrimental to scientific progress than false positive results pertaining to understudied genes.

Second, I did not track how the GSK study relates to this concern--the study found that published claims about well-studied genes were not especially predictive of clinical trial success, but this only seems to be of concern if there is some contrast showing that published claims of understudied genes are consistently more accurate (ref 17 also seems out of place here, as it is an opinion piece musing about the need for more stringent significance thresholds in medical research and doesn't contribute any empirical evidence supporting the claims made in line 44).

2. Lines 168-169: It seems likely that many of the recent early-stage publications produced by larger teams are coming from GWAS/TWAS studies where teams are more or less agnostically searching for association signals in any and all genes. This shouldn't be particularly surprising, since the number of authors correlates with GWAS sample size and thus also correlates with power to detect novel associations, which presumably yields additional hits in relatively understudied genes. To me, this suggests the most parsimonious answer to "how do we encourage early-stage research on understudied genes?" is simply "fund/collaborate on bigger GWAS!", which seems somewhat antithetical to the authors' goal of increasing innovation in the life sciences. 

See also lines 189-192—"since the mid-2000s" aligns with when the first successful GWAS were published, so I'd expect much of what the authors are interpreting as recent bibliometric recognition/rewards for early-stage research could alternatively be interpreted as bibliometric recognition for GWAS as a methodology and subdiscipline of genomics (this is even alluded to in lines 333-337, where the NIH-funded organizations with the strongest enrichment for early-stage research are solely driven by GWAS-related publications, and lines 416-417, describing the initial discoveries of FTO and C9orf72 via GWAS). 

Ultimately, I'm curious if these results and interpretations change when GWAS publications are excluded from the analyses. Struck et al. (2018) [https://www.ncbi.nlm.nih.gov/pmc/articles/PMC6090631/] is an essential reference here, as they show that genes newly implicated by GWAS in complex disease do experience additional publications compared to control genes with similar publication histories. (The trend described in lines 400-420 was previously characterized by Struck et al., where a subset of recently-highlighted genes discovered via GWAS have received extraordinary attention in the literature in a short period of time).

3. I'd also like to see a more nuanced discussion of how pleiotropy fits into this study--a gene may already be established as "well-studied" for its association with a particular disease, but discovery of a novel connection between that gene and another disease can be just as impactful as an association between a novel gene and that disease (if not more so because we likely already know much about that well-studied gene's biology so it is potentially better poised for translational applications).

Minor comments:

Line 29: Consider also including Pfeiffer & Hoffman (2007) in this list [https://www.ncbi.nlm.nih.gov/pmc/articles/PMC1924584/].

Lines 84-86: apologies for being pedantic, but this isn't strictly true--the "full space of the possibly explorable entities" has only been truly available after the recently published telomere-to-telomere sequencing paper, which found an additional ~2,200 paralogous gene copies and 115 protein-coding genes (https://www.biorxiv.org/content/10.1101/2021.05.26.445798v1).

Line 166: delete "is"

Lines 294-298: The finding that HHMI-supported publications are enriched for early-stage research lends itself nicely to the argument that funding agencies can pursue high-risk, potentially high-reward research by "funding people not projects" (https://www.nature.com/articles/477529a). This argument would interface nicely with the discussion in lines 482-485, where the "people not projects" ethos might be particularly productive and valuable if targeted towards early-career researchers.

Lines 309-310: "strong support" and "avoidance" imposes an unnecessary judgement--better to just say the focus on early-stage research among domain-specific funding agencies can differ across countries and can change over time as those agencies reassess and reprioritize the areas of research they will pursue in any given year. (Also line 493--"pursuit or avoidance of innovation" implies NCI-funded research is not innovative, which might be read as an affront to a massive sector of this manuscript's potential audience).

Reviewer #3:

I am happy with the current shape of the manuscript. I especially like the clear formulation and testing of hypotheses.

---

## [Editor Report · Decision Letter 2]

21 Dec 2021

Dear Thomas,

On behalf of my colleagues and the Academic Editor, Marcus Munafò, I'm pleased to say that we can in principle accept your Meta-Research Article "The characteristics of early-stage research into human genes are substantially different from subsequent research" for publication in PLOS Biology, provided you address any remaining formatting and reporting issues. These will be detailed in an email that will follow this letter and that you will usually receive within 2-3 business days, during which time no action is required from you. Please note that we will not be able to formally accept your manuscript and schedule it for publication until you have any requested changes.

PRESS: We frequently collaborate with press offices. If your institution or institutions have a press office, please notify them about your upcoming paper at this point, to enable them to help maximise its impact. If the press office is planning to promote your findings, we would be grateful if they could coordinate with biologypress@plos.org. If you have not yet opted out of the early version process, we ask that you notify us immediately of any press plans so that we may do so on your behalf.

Sincerely, 

Roli

Roland G Roberts, PhD 

Senior Editor 

PLOS Biology

rroberts@plos.org